# Transcriptional down-regulation of *ccr5* in a subset of HIV+ controllers and their family members

**Elena Gonzalo-Gil[1], Patrick B Rapuano[1], Uchenna Ikediobi[1], Rebecca Leibowitz[1], Sameet Mehta[2], Ayse K Coskun[1], J Zachary Porterfield[1], Teagan D Lampkin[3], Vincent C Marconi[4], David Rimland[4], Bruce D Walker[5], Steven Deeks[6,7], Richard E Sutton[1]***

[1]Section of Infectious Diseases, Department of Internal Medicine, Yale University School of Medicine, New Haven, United States; [2]Yale Center for Genome Analysis Bioinformatics group, Yale University School of Medicine, New Haven, United States; [3]Infectious Diseases Section, Dallas VA Medical Center, Dallas, United States; [4]Atlanta VA Medical Center, Emory University School of Medicine, Atlanta, United States; [5]Ragon Institute of MGH, MIT and Harvard University, Cambridge, United States; [6]Department of Medicine, University of California San Francisco, San Francisco, United States; [7]Department of Epidemiology and Biostatistics, University of California San Francisco, San Francisco, United States

**Abstract** HIV +Elite and Viremic controllers (EC/VCs) are able to control virus infection, perhaps because of host genetic determinants. We identified 16% (21 of 131) EC/VCs with CD4 +T cells with resistance specific to R5-tropic HIV, reversed after introduction of *ccr5*. R5 resistance was not observed in macrophages and depended upon the method of T cell activation. CD4 +T cells of these EC/VCs had lower *ccr2* and *ccr5* RNA levels, reduced CCR2 and CCR5 cell-surface expression, and decreased levels of secreted chemokines. T cells had no changes in chemokine receptor mRNA half-life but instead had lower levels of active transcription of *ccr2* and *ccr5*, despite having more accessible chromatin by ATAC-seq. Other nearby genes were also down-regulated, over a region of ~500 kb on chromosome 3p21. This same R5 resistance phenotype was observed in family members of an index VC, also associated with *ccr2*/*ccr5* down-regulation, suggesting that the phenotype is heritable.
DOI: https://doi.org/10.7554/eLife.44360.001

*For correspondence:
richard.sutton@yale.edu

Competing interests: The authors declare that no competing interests exist.

## Introduction

Human immunodeficiency virus type 1 (HIV-1) is pandemic, with more than 36 million people infected world-wide. Anti-retroviral therapy (ART) is a mainstay of treatment, but once therapy is stopped or drug resistance develops, viral rebound occurs within weeks and CD4 +T cell counts decline (*Holkmann Olsen et al., 2007*). A small population of HIV-infected individuals termed elite controllers (ECs) and viremic controllers (VCs), however, are able to control viral replication (plasma viral load, VL <50 [ECs] or 50 < VL < 2000 [VCs] for at least 6–12 months) in the absence of ART by a mechanism that is not fully elucidated (*Deeks and Walker, 2007*; *Gonzalo-Gil et al., 2017*; *Lambotte et al., 2005*). EC/VCs are considered examples of 'functional' cures, in which virus is not fully eradicated and yet for the most part the patient does not develop immune dysfunction over time. The clinical status of most EC/VCs cannot be explained by defective HIV particles or genomes (*Wang et al., 2002*; *Blankson et al., 2007*). Rather, these individuals appear to have an intrinsic ability to control HIV infection, perhaps because of host genetic determinants. A genome-wide

association study (GWAS) identified certain human leukocyte antigens (HLA)-B and HLA-C alleles that are associated with viral control in ECs (*Pereyra et al., 2010*). However, these protective alleles only accounted for ~20% of the effect, suggesting that there are other mechanisms responsible for the suppressed viral loads in EC/VCs. Identifying novel mechanisms involved in HIV control is paramount to HIV research and the cure agenda.

C-X-C chemokine receptor 4 (CXCR4) and C-C chemokine receptor 5 (CCR5) serve as co-receptor for X4-tropic and R5-tropic HIV-1 entry into CD4 +T cells, respectively, and CCR5 is essential for sexual transmission of HIV (*Feng et al., 1996*). The presence of the *CCR5* delta 32 (Δ32*CCR5*) allele confers protection against seroconversion, with homozygotes being completely resistant to infection via mucosal routes (*Liu et al., 1996*; *Samson et al., 1996*). There is, however, no evidence that Δ32*CCR5* ± is associated with EC/VC phenotype. Conflicting results have been obtained regarding the susceptibility of EC/VC CD4 +T cells to HIV infection in vitro. Activated CD4 +T cells from EC/VCs have been shown to be susceptible to both R5- and X4-tropic HIV (*Blankson et al., 2007*; *Lamine et al., 2007*) but opposite results have also been reported, with CD4 +T cells of EC/VCs being resistant to HIV (*Chen et al., 2011*; *Sáez-Cirión et al., 2011*; *Walker et al., 2015*; *Julg et al., 2010*).

Previously we had observed that three of roughly a dozen ECs tested had CD4 +T cells with intrinsic resistance to R5 virus, due to increased chemokine gene expression (*Walker et al., 2015*). To extend those findings and to determine whether R5 resistance is a consequence of a transcriptional mechanism and if there is a hereditary basis associated with the phenotype, we analyzed the in vitro susceptibility to HIV of purified CD4 +T cells from 131 EC/VCs, along with normal, healthy donors. Here we report that a subset of EC/VCs have resistance to HIV, specific to R5-tropic virus. For these subjects, however, the resistance phenotype was due to lower levels of CCR5, at both the RNA and protein levels, and was likely due to reduced active transcription of *ccr5*, despite highly accessible chromatin. The fact that CD4 +T cells from multiple family members of an index VC had a similar phenotype and also down-regulation of *ccr5* suggests that the phenotype is hereditary in nature.

## Results

### Clinical characteristics of EC/VC cohort

The total number of EC/VCs studied was 131, with a majority coming from the UCSF SCOPE cohort. Forty-four percent (58/131) were ECs, with 56% (73/131) being VCs (See *Supplementary file 1*). The year of initial HIV diagnosis or likely exposure ranged from 1980 to 2014, and subjects were 48 ± 12 years old (mean ±SD, range of 19 to 79 years), the majority being men (78.62%). CD4 +T cell count at time of enrollment was 689 ± 358 (mean ±SD). Most had never received ART except under the circumstances of pregnancy or malignancy (*Supplementary file 1*). Although occasional viral blips were observed, none of the EC/VCs ever lost virologic control necessitating ART. A number of subjects (54/125) had documented protective HLA alleles, being 32.06% HLA-B*57:03, 25.95% HLA-B*57:01, 22.9% Cw*08:02, 10.69% B*14:02, 4.58% HLA-B*27:05, and 3.05% B*52:01.

### In vitro CD4 +T cell intrinsic resistance specifically to R5-tropic virus in a subset of HIV +EC/VCs

To determine whether T cells of EC/VCs were resistant to X4- or R5-tropic virus in vitro, we activated CD4 +T cells from 131 EC/VC and 35 Ctrl, and then infected them overnight using single cycle HIV encoding YFP and pseudotyped with either X4, R5, or VSV G glycoprotein and analyzed cells by flow cytometry 72 hr later. We observed relative resistance to R5-tropic HIV in CD4 +T cells from EC/VCs (% cells eYFP+: EC/VC 0.99 ± 0.79) compared to Ctrl (1.22 ± 0.66; p=0.01; *Figure 1—figure supplement 1A*, left panel). In contrast, we saw equal susceptibility to X4-tropic HIV (Ctrl 3.08 ± 1.32; EC/VCs 3.33 ± 1.91) and VSV G pseudoviral particles among the groups (Ctrl 34.8 ± 9.36; EC/VCs 30.66 ± 11.22; *Figure 1—figure supplement 1B*). Post-hoc analysis identified 16% of EC/VCs (21 of 131 analyzed, termed ECr/VCr) with resistance specific to R5-tropic HIV, compared to remaining EC/VC subjects and healthy Ctrl, with no resistance observed (% cells YFP+: Ctrl 1.22 ± 0.66; EC/VC 1.2 ± 0.77; ECr/VCr 0.2 ± 0.07; p<0.0001; *Figure 1—figure supplement 1A*, right panel), pointing to an early block of infection in a subset of EC/VCs. These data confirmed that the phenotype was

specific to EC/VC, not observed in Ctrl. To confirm the R5 resistance phenotype, we then selected ECr/VCr samples for further study, based upon % eYFP +cells being lower than any value in Ctrl group. We retested these ECr/VCr samples prospectively in at least triplicate, using two R5-tropic envelopes, in comparison to a subset of EC/VC (n = 38, selected based upon sample availability and representativeness of the population from the initial test) and Ctrl (n = 35). Our results redemonstrated R5 resistance, as manifested as a 5-fold reduction in CD4 +T cell susceptibility to YU2-pseudotyped virus, on average, in ECr/VCr compared to remaining EC/VC and Ctrl (*Figure 1A*, % cells eYFP+: Ctrl 1.05 ± 0.81; EC/VC 1.09 ± 0.75; ECr/VCr 0.20 ± 0.16; p<0.0001). Similar results were

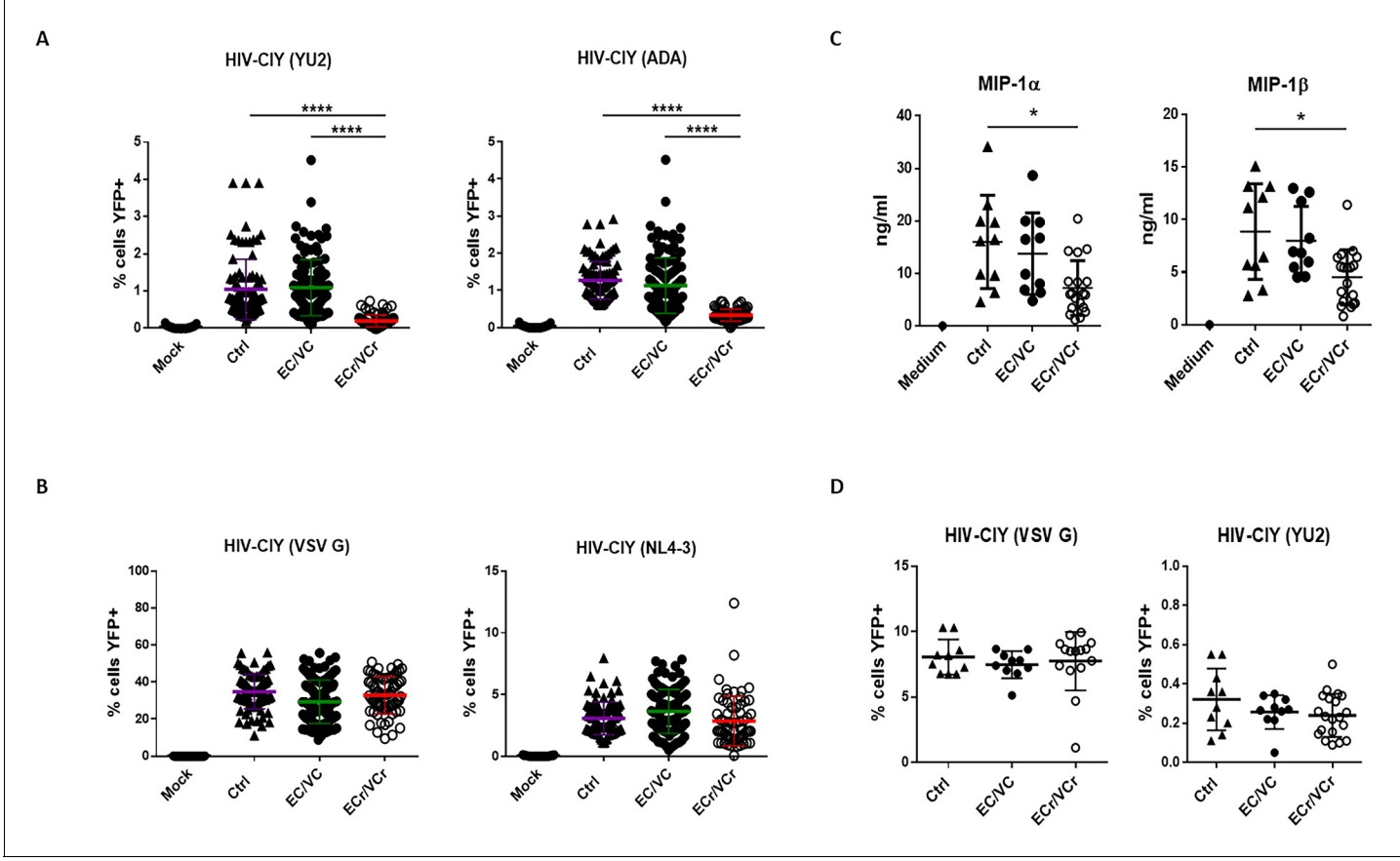

**Figure 1.** CD4 +T cell resistance to infection in prospective single cycle assay, specific to R5-tropic viruses in a subset of EC/VCs. (**A**) Five-fold resistance to R5-tropic viruses in 16% of EC/VC (ECr/VCr) infected using replication defective HIV-cycT1-IRES-eYFP (CIY) with R5-tropic envelopes YU2 and ADA. A > 95% power was determined based on comparisons of means using PASS statistical software between ECr/VCr and all other groups (Ctrl and EC/VC). (**B**) Equivalent susceptibility to both X4-tropic (NL4-3) and VSV G pseudoviral particles in ECr/VCr. A and B are pooled results from different experiments with samples tested at least in triplicate (Ctrl n = 35, EC/VC n = 38, representative from the initial population (*Figure 1—figure supplement 1*) and selected based upon specimen availability, and ECr/VCr (n = 21). (**C**) Comparable levels of chemokines (MIP-1α and MIP-1β) in cell culture supernatants from activated CD4 +T cells, measured by ELISA. (**D**) CD4 +T cells from Ctrl were exposed to cell culture supernatants from activated T cells of Ctrl and EC/VC with or without the resistance phenotype, in the presence of HIV particles pseudotyped with YU2 or VSV G. C and D are pooled results from different experiments with n = 10 (Ctrl and EC/VCs) and n = 21 (ECr/VCr). Shown are individual values with Means ± Standard Deviation (SD). Data were analyzed by using the Kruskal-Wallis test and Dunn's multiple-comparison test. *p<0.05; ****p<0.0001.

DOI: https://doi.org/10.7554/eLife.44360.002

The following figure supplements are available for figure 1:

**Figure supplement 1.** Initial testing showing CD4 +T cell resistance to infection in single-cycle pseudotyping assay, specific to R5-tropic viruses, in a subset of EC/VCs.
DOI: https://doi.org/10.7554/eLife.44360.003

**Figure supplement 2.** CD4 +T cell resistance to infection in single-cycle pseudotyping assays in 21 ECr/VCr and multi-cycle infection in a subset of ECr/VCr, Ctrl and EC/VC.
DOI: https://doi.org/10.7554/eLife.44360.004

observed using ADA-pseudotyped virus (% cells YFP+: Ctrl 1.27 ± 0.5; EC/VC 1.13 ± 0.75; ECr/VCr 0.34 ± 0.16; p<0.0001, *Figure 1A*). Similar to the post-hoc analysis, in this prospective testing we observed equal susceptibility to X4-tropic and VSV-G-pseudotyped HIV particles in activated CD4 +T cells from ECr/VCr compared to EC/VC without the phenotype and Ctrl (*Figure 1B*). In multiple cases, based upon sample and subject availability, we retested ECr/VCr CD4 +T cells isolated from independent, separate blood draws and observed consistent results (i.e., R5 resistance was seen repeatedly, not just on a single blood draw). Taken together, these data identify a subset of EC/VCs with intrinsic, reproducible resistance specific to R5-tropic virus in T cells, a phenotype only observed in EC/VC. From the 21 EC/VCs with the resistance phenotype, 43% were ECs (9/21) and 57% VCs (12/21). *Figure 1—figure supplement 2A* shows virus infectivity data for all 21 ECr/VCr, with *Figure 1—figure supplement 2B* demonstrating absence of correlation between R5 and X4 and R5 and VSV G susceptibility.

We next analyzed whether any clinical characteristics (VL, CD4 +T cell count, and age) were associated with the R5 resistance phenotype in the EC/VC population. Comparable VLs and CD4 +T cell counts were observed in both groups (*Figure 1—figure supplement 1C*). However, ECr/VCr were significantly younger than EC/VC (43 ± 14 vs 49 ± 12 years; p=0.047; *Figure 1—figure supplement 1C*). Analyzed by gender, most of the subjects in both groups were men (EC/VC 78% or 85/109% and 86% or 18/21 in ECr/VCr).

To investigate whether this resistance was associated with increased levels of chemokines or other soluble factors, which could block viral entry by competitively binding to the chemokine co-receptor CCR5 (*Paxton et al., 1996*; *Saha et al., 1998*), chemokine levels were quantified in cell culture supernatants from activated CD4 +T cells. We selected samples from each group (Ctrl and EC/VCs based upon specimen availability and representative from the initial testing) and compared with ECr/VCr (n = 21). oup (Ctrl and EC/VCs basing on sample availability and representativeness from the initCD4 +T cells from ECr/VCr, however, had decreased levels of secreted MIP-1α and MIP-1β, compared to the other groups (*Figure 1C*), which was statistically significant compared to Ctrl (MIP-1α: Ctrl 16 ± 9.42 vs ECr/VCr 7.24 ± 5.21 ng/ml; p=0.048 and MIP-1β: ECr/VCr 4.52 ± 2.61 vs Ctrl 9.12 ± 4.73 ng/ml; p=0.01). Additionally, we performed media transfer experiments to explore whether other factors elaborated by activated CD4 +T cells were responsible to the resistance phenotype in this ECr/VCr subset. Our results revealed comparable T cell susceptibility to infection in ECr/VCr and EC/VCs without the phenotype (*Figure 1D*), suggesting that the culture supernatants did not contain soluble factors that could confer resistance to R5-tropic virus in the ECr/VCrs.

Previous reports have suggested that expression of HLA-B*27/HLA-B*57 and other specific HLA alleles can account for some of the controller phenotype. We examined whether the presence of protective HLA alleles was associated with viral control in ECr/VCr subset. Of the 16% of ECr/VCr with the R5 resistance phenotype, only five individuals (5/19 or 26.3%) had documented protective alleles, with four of them being HLA-B*57 positive and only one HLA-B*27. Analyzing the remaining EC/VC, the percentage was higher, with 46% (49/106) of them having protective HLA alleles. Although this difference in frequency of protective alleles was not significant (p=0.086) due to the low number of ECr/VCr, these data confirm that protective alleles were not more frequent in ECr/VCr.

We next investigated whether the ECr/VCr CD4 +T cells were also relatively resistant to replication-competent virus. Activated CD4 +T cells from EC/VC, ECr/VCr (based upon prior experiments) and Ctrl (n = 2 per group, tested in triplicate, selected based upon cell availability) were infected with X4- and R5-tropic viruses, at low MOI. Viruses were prepared in 293 T cells by co-transfection with VSV G expression plasmid to facilitate the first round of replication. Replication of NL4-3 (X4) and BaL (R5) was quantified using TZMbl cells as a reporter, measuring firefly luciferase activity over a period of 3 weeks. We observed significantly reduced replication of BaL in ECr/VCr, compared to EC/VC and Ctrl over the 21 days analyzed (Mean ±SD Area Under Curve [AUC] R5: Ctrl 177828 ± 53736; EC/VC 125548 ± 31577; ECr/VCr 62006 ± 4179; *Figure 1—figure supplement 2C*, right panel). The absence of differences in viral replication at day three post-infection may be explained by the addition of VSV G as described above. Infection using NL4-3 also showed significant resistance in all EC/VC (AUCs: Ctrl 19679 ± 12897; EC/VC 5880 ± 1319; ECr/VCr 2125 ± 60.1, *Figure 1—figure supplement 2C*, left panel). The fact that EC/VC (with or without the R5-resistance phenotype) had reduced infectivity, with virtual absence of X4 replication in ECr/VCr, suggests a more complex mechanism of virologic resistance that should be further explored.

# RNA-Seq identifies several genes down-regulated in EC/VC with R5-tropic resistance

To further investigate the mechanism of R5-tropic resistance in early infection, we next performed RNA-Seq to identify genes that were significantly up- or down-regulated in activated CD4 +T cells from ECr/VCr compared to Ctrl. We examined RNA levels in activated T cells because those are the cells in which we observed the R5 resistance phenotype (unactivated T cells are extremely difficult to infect). Several of the differentially expressed genes were located on chromosome 3 (chr 3), including *ccr1*, *ccr2*, and *ccr5*, which were significantly down-regulated in ECr/VCr (corrected p values=0.005). To quantify mRNA levels of these genes in ECr/VCr, we performed RT-qPCR in ECr/VCr, and compared results to remaining EC/VCs and Ctrl. These data confirmed a 7-fold decreased expression in *ccr2* mRNA levels, on average, in T cells of ECr/VCr (0.13 ± 0.09) compared to those of EC/VC without the resistance phenotype (0.89 ± 0.41; p<0.0001) and Ctrl (0.91 ± 0.72; p<0.0001; *Figure 2A*). Similarly, we observed down-regulation of *ccr5* RNA in T cells of ECr/VCr (0.076 ± 0.047; 9-fold decrease on average) compared to those of the other groups (EC/VC 0.79 ± 0.63 and Ctrl 0.68 ± 0.63; p<0.0001, *Figure 2A*).

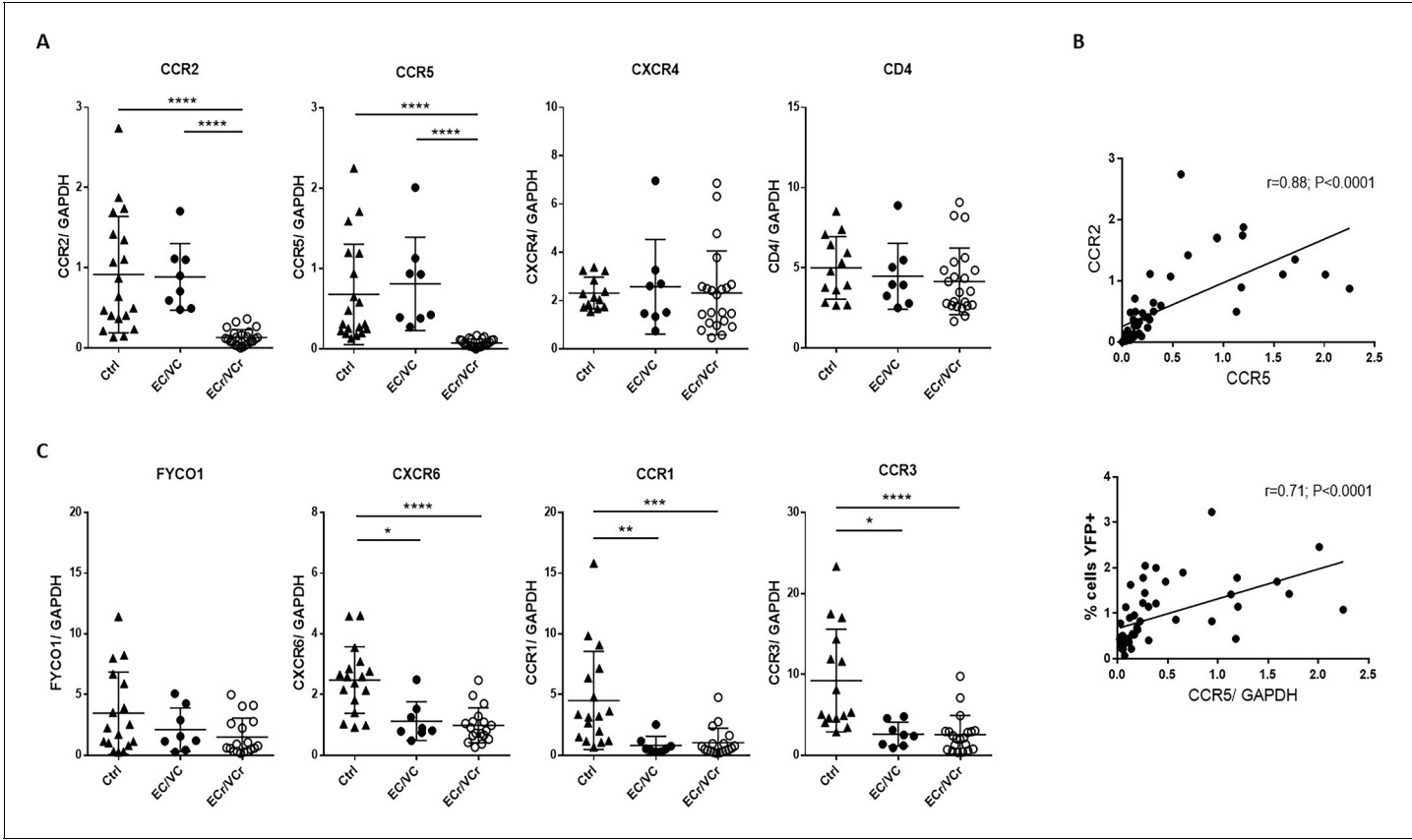

**Figure 2.** Decreased mRNA levels of several chromosomal three genes in ECr/VCrs. (**A**) Decreased *ccr2/ccr5* RNA levels in activated CD4 +T cells from EC/VCs with the resistance phenotype, with comparable *cxcr4* and *cd4* RNA levels in all groups. Shown are individual values with Means ± SD. Pooled results from different experiments are shown with representative samples per group, n = 19 (Ctrl), n = 8 (EC/VC) and n = 21 (ECr/VCr) per group. (**B**) Positive correlation between *ccr2* and *ccr5* RNA levels in activated CD4 +T cells. *ccr5* RNA levels positively correlated with % of YFP +infected cells by single cycle assay using R5-tropic viruses but not with *cd4* or *cxcr4* (*Figure 2—figure supplement 1A*). (**C**) Decreased RNA levels in multiple chromosomal 3p21 genes in T cells of HIV +infected individuals (*Figure 2—figure supplement 1C*). Statistical analysis performed using Kruskal-Wallis test and Dunn's multiple-comparison test. r value calculated using the non-parametric Spearman correlation test. Graphs show individual values with Means ± SD. *p<0.05; **p<0.01; ***p<0.001; ****p<0.0001.

DOI: https://doi.org/10.7554/eLife.44360.005

The following figure supplement is available for figure 2:

**Figure supplement 1.** Correlations and fold-change in RNA levels in ECr/VCr.

DOI: https://doi.org/10.7554/eLife.44360.006

Conversely, we did not observe significant differences between groups in *cxcr4* and *cd4* RNA levels (*Figure 2A*). Interestingly, *ccr5* RNA highly correlated with *ccr2* RNA levels (r = 0.88; p<0.0001, *Figure 2B*), suggesting a common regulatory mechanism for both genes in all subjects. Moreover, *ccr5* mRNA levels were positively correlated with transduction by R5-tropic virus (r = 0.71; p<0.0001, *Figure 2B*), indicating that subjects whose CD4 +T cells were more resistant to R5-tropic virus had lower *ccr5* mRNA expression. However, there was no correlation between *ccr2* and *cd4* or *cxcr4* RNA levels, nor between *ccr5* and *cd4* or *cxcr4* RNA levels (*Figure 2—figure supplement 1A*).

Because several genes near *ccr2/ccr5* appeared to be down-regulated, we analyzed levels of gene expression both centromeric and telomeric of that region. We observed down-regulation in several genes in that locus of 3p21, including *fyco1*, *cxcr6*, *ccr1*, and *ccr3* in all HIV +infected groups (EC/VC, and ECr/VCr) compared to Ctrl, although only EC/VC and ECr/VCr groups reached statistical significance compared to Ctrl (*Figure 2C*). Taken together, these data point towards RNA down-regulation involving a region of approximately 500 Kb, surrounding *ccr2/ccr5* in EC/VC (*Figure 2—figure supplement 1C*), with a more profound decrease of *ccr2/ccr5* specifically in ECr/VCr, not observed in the remaining EC/VCs.

*LOC102724291* is a poorly characterized long non-coding RNA (lncRNA) of unknown function, present on chr3, antisense to *ccr5* and *ccr2*. To ascertain if *loc102724291* was involved in *ccr2/ccr5* RNA down-regulation, we quantified its expression in CD4 +T cells by RT-qPCR. Comparable levels were observed between ECr/VCr and EC/VCs without the phenotype using a primer pair within exons 1 and 2. We did observe, however, lower lncRNA levels using a primer pair within exon 3, within intron 2 of *ccr5* (*Figure 2—figure supplement 1D*), in CD4 +T cells of ECr/VCr compared to other groups. The absence of a negative correlation between *ccr2* or *ccr5* and *loc102724291* makes it unlikely that an antisense effect from this lncRNA is responsible for the down-regulation of *ccr2/ccr5* in ECr/VCr and is more consistent with *loc102724291* also being down-regulated by a more global mechanism, similar to other genes in the region.

## Lower CCR2 and CCR5 surface expression in EC/VC with the resistance phenotype

We confirmed the activation status of CD4 +T cells by analyzing CD69 and CD25 up-regulation by flow cytometry, cell-surface markers of early and late cell activation, respectively, after CD4 +T cell activation with aCD3/CD28 for three days. Results showed a strong late activation of CD4 +T cells in all groups, with comparable CD25 levels in ECr/VCr and remaining EC/VC and Ctrl groups (*Figure 3A*). However, we observed lower levels of CD69 (%+) in activated CD4 +T cells from ECr/VCr (22.26 ± 6.54) compared to EC/VC without the resistance phenotype (30.59 ± 5.15; p=0.011) and Ctrl (32.12 ± 4.15; p=0.0003; *Figure 3A*).

It is important to note that we infected the T cells 72 hr after activation, not at 24 hr, and that the resistance phenotype was specific to R5 virus, with equal susceptibility to X4- and VSV G-pseudo-typed particles. In fact, positive correlations were observed between *ccr5* mRNA or cell surface expression levels and % CD69 +cells, confirming that lower *ccr5* expression, but not *cxcr4*, was observed in CD4 +T cells with lower levels of the early activation marker (*Figure 2—figure supplement 1B*). In select samples we also analyzed the percentage of memory T cells after anti-CD3/CD28 co-stimulation. Results showed a high percentage of CD45RO + memory T cells of between 60–80%, and low percentage of naïve CD45RA + T cells (~10%), with no differences between groups (*Figure 3B*). These data, taken together, confirm efficient activation of CD4 +T cells in all subject groups studied, with a high percentage of memory T cells after activation.

To determine whether the resistance phenotype was associated with an alteration in the expression of CCR2 and CCR5, cell surface levels were quantified by flow cytometry in activated CD4 +T cells. Our results revealed lower surface expression of CCR2 in ECr/VCr (15.5 ± 10.17 %+) compared to other EC/VCs (22.99 ± 11.04; p=0.021) and Ctrl (23.64 ± 9.13; p=0.023; *Figure 3C*, CD3/28 panel. *Figure 3—figure supplement 1* shows individual flow cytometric histograms, comparing cell surface expression of Ctrl and all 21 ECr/VCr). Similarly, differences in CCR5 expression also reached significance, being lower in ECr/VCr (21.39 ± 13.65) than in other EC/VC (37.99 ± 12.3; p=0.005) or Ctrl (36.08 ± 10.29; p=0.003; *Figure 3C*, CD3/28 panel. *Figure 3—figure supplement 2* shows individual flow cytometric histograms comparing cell surface expression between Ctrl and all 21 ECr/VCr). Similar results were observed analyzing the data as MFI, with lower CCR2 and CCR5 in ECr/VCr compared to the other groups (*Figure 3E*). Interestingly, we observed a positive correlation between

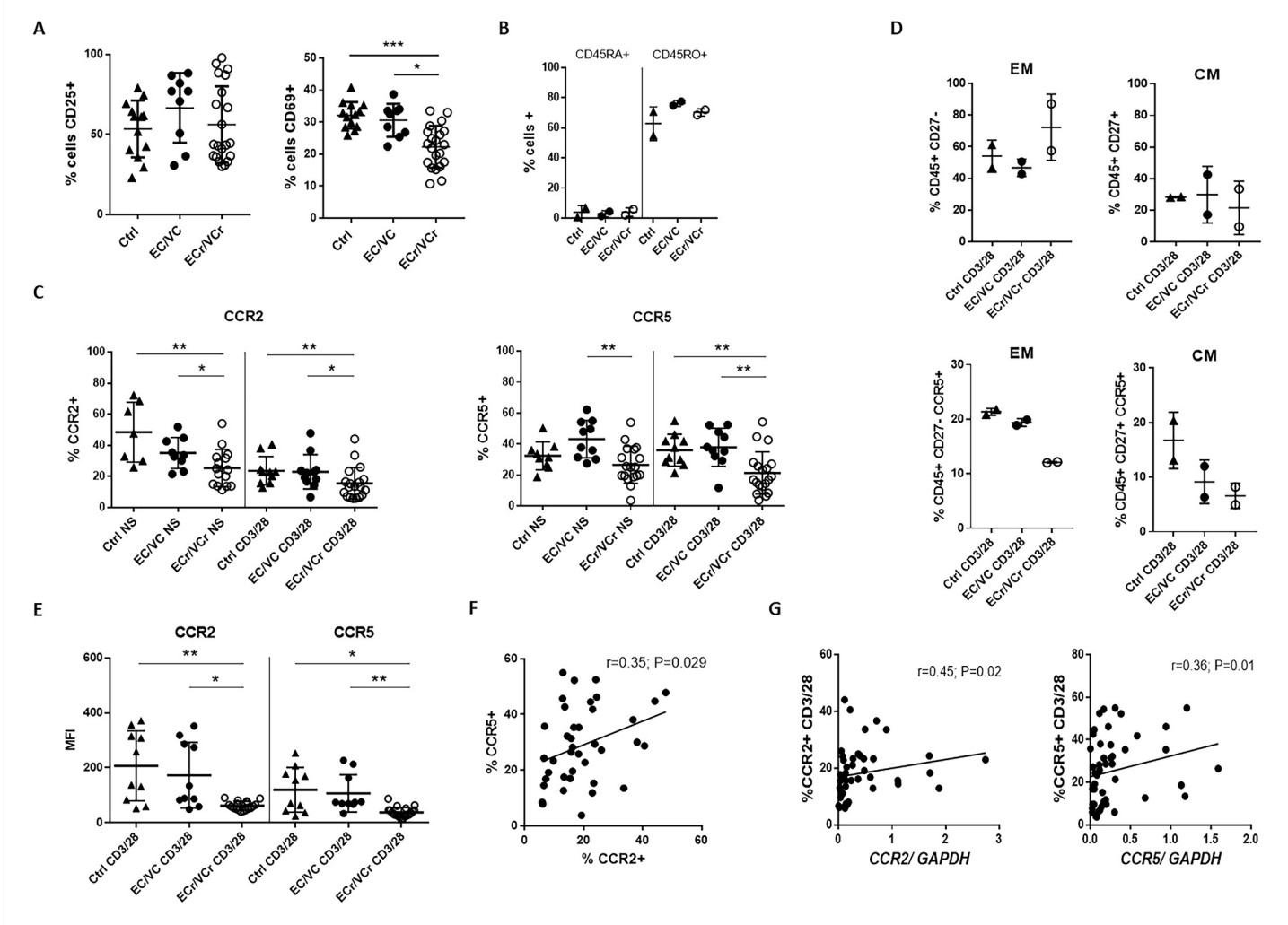

**Figure 3.** Lower proliferative responses and CCR2 and CCR5 cell surface levels in activated CD4 +T cells from ECr/VCrs. (A) Reduced CD69, but not CD25 levels in activated CD4 +T cells from ECr/VCrs. Graph shows representative data N = 13 (Ctrl), n = 9 (EC/VC) and n = 21 (ECr/VCr). (B) Comparable frequencies of naïve CD45RA + and memory CD45RO + T cells after anti-CD3/CD28 activation between groups (n = 2 per group). (C) CCR5 and CCR2 cell surface levels measured by flow cytometry are reduced in freshly thawed (NS, non-stimulated) and activated CD4 +T cells (anti-CD3/28) from ECr/VCr. (D) Percentages of CCR5 +in effector memory (EM) and central memory (CM) compartments of activated CD4 +T cells (n = 2 per group). (E) Reduced CCR2 and CCR5 cell surface levels, expressed as MFI, in activated (anti-CD3/28) CD4 +T cells from ECr/VCr. Data in D-E shown pooled results from different experiments with n = 10 (Ctrl and EC/VC) and n = 19 (ECr/VCr). (F) Positive correlation between CCR2 and CCR5 cell surface levels. (G) Positive correlation observed between *ccr2/ccr5* RNA levels and cell surface expression. Values obtained using the non-parametric Spearman correlation test. *p<0.05.

DOI: https://doi.org/10.7554/eLife.44360.007

The following figure supplements are available for figure 3:

**Figure supplement 1.** Flow cytometric histograms showing CCR2 +T cells in all 21 ECr/VCr, compared to 10 Ctrl, with isotype control staining in blue (open histograms).
DOI: https://doi.org/10.7554/eLife.44360.008

**Figure supplement 2.** Flow cytometric histograms showing CCR5 +T cells in all 21 ECr/VCr, compared to 10 Ctrl, with isotype control staining in blue (open histograms).
DOI: https://doi.org/10.7554/eLife.44360.009

CCR2 and CCR5 surface expression (r = 0.35; p=0.029; *Figure 3F*). We next investigated whether there were differences in surface expression levels in non-stimulated (NS) CD4 +T cells. Similarly, CCR2 and CCR5 expression levels in NS CD4 +T cells were significantly lower in ECr/VCr, compared to EC/VC without the resistance phenotype (*Figure 3C*). Since most CCR5 +CD4+T cells show an

effector memory phenotype (EM, defined as CD45RO+/CD27-), and these T cells may be more easily infected by R5-tropic virus, we investigated whether CCR5 levels were lower in Effector Memory (EM) from ECr/VCr. Our results, however, demonstrated lower CCR5 expression in both EM and central memory T cells (CM, defined as CD45RO+/CD27+; *Figure 3D*) in ECr/VCr, compared to Ctrl and EC/VC. In addition, percentages of EM trended higher in ECr/VCr T cells compared to other groups, suggesting that the R5-resistance phenotype is not due to a lower percentage of EM T cells.

We then investigated whether activated CD4 +T cells from individuals with lower *ccr2/ccr5* RNA levels also had lower surface expression of both CCR2 and CCR5. We saw a positive correlation between CCR2 protein expression and *ccr2* RNA levels (r = 0.45; p=0.02, *Figure 3G*). Similar results were observed with CCR5 (r = 0.36; p=0.01), suggesting that down-regulation of *ccr5* RNA was responsible for lower cell surface expression and consequent resistance to R5 virus in ECr/VCr CD4 +T cells.

## Increased susceptibility to R5-tropic virus infection in activated CD4 +T cells after overexpression of CCR5

To confirm that the R5-tropic resistance in ECr/VCr CD4 +T cells was due to down-regulation of CCR5, activated CD4 +T cells from ECr/VCr, EC/VC, and Ctrl were infected with R5-tropic pseudotyped HIV particles after cell transduction using pan-tropic pseudotyped viral particles encoding both CCR5 and eYFP. First, we observed an increase in the percentage of CCR5 +cells in ECr/VCr transduced with pHIV-CCR5-IRES-YFP (VSV G) (8.09 ± 3.86%), compared to vector encoding YFP alone (3.16 ± 1.14%; p=0.032, *Figure 4A*). These CD4 +T cells were more susceptible than those of Ctrl to subsequent infection using two different R5-tropic viruses (YU2: Ctrl 1.15 ± 0.05% vs ECr/VCr 2.74 ± 1.11%; p=0.009; ADA: Ctrl 0.70 ± 0.04% vs ECr/VCr 1.79 ± 1.3%; p=0.008).

Also, and more interestingly, higher susceptibility was observed in ECr/VCr than EC/VC (for YU2: EC/VC 1.56 ± 0.39% vs ECr/VCr 2.47 ± 0.51%; p=0.03, *Figure 4A*). We did not, however, observe any differences when we used VSV G-pseudotyped viral particles, confirming that the observed R5-resistance phenotype in ECr/VCr was in fact due to decreased cell surface expression of CCR5.

To determine whether this R5-resistance phenotype was observed in other circulating mononuclear cells, macrophages derived from monocytes (MDMs) were infected using pseudotyped lentiviral particles and analyzed by flow cytometry (*Figure 4B*). We observed comparable R5 susceptibility in MDMs from ECr/VCr and remaining EC/VCs. We next analyzed *ccr5* and *ccr2* RNA expression levels in MDMs from EC/VCs, and equivalent levels were present in all groups (*Figure 4C*). Similarly, the percentages of CCR5 +and CCR2 +in CD14+cells were comparable between groups (*Figure 4D*), suggesting that the R5-tropic resistance phenotype and *ccr2/ccr5* down-regulation observed in a subset of EC/VCs were specific to activated CD4 +T cells.

Other investigators have attempted to determine with limited success whether EC/VC CD4 +T cells are resistant to infection in vitro. To ascertain whether the conflicting results are a consequence of varying experimental conditions or clinical characteristics of the EC/VCs, we activated CD4 +T cells using PHA or PMA/ionomycin from a representative number of samples from different groups, and T cells then infected with pseudotyped viral particles. We observed comparable CD4 +T cell susceptibility to X4- and VSV G-pseudotyped particles in EC/VCs (*Figure 5A*).

Additionally, we did not observe significant differences in R5-tropic virus susceptibility of EC/VC CD4 +T cells after PMA/ionomycin stimulation (*Figure 5B*), although T cell susceptibility trended lower in ECr/VCr compared to Ctrl and EC/VCs after PHA stimulation. Next, we analyzed *ccr2* and *ccr5* transcript levels by qPCR in the same samples after both experimental conditions and results were comparable between groups (*Figure 5C*). Similarly, no differences were found in CCR5 cell surface expression between groups after both non-specific stimulations (*Figure 5D*). Our data thus suggest that the R5-tropic resistance phenotype in ECr/VCr is limited to CD4 +T cells activated by anti-CD3/CD28 co-stimulation, which in vitro is the most physiological method of stimulation, short of using cognate antigen and antigen presenting cells.

## Frequencies of Δ32*CCR5* and promoter polymorphism in EC/VC with resistant phenotype

In order to exclude the possibility that the observed R5-tropic resistance in ECr/VCr was due to the *ccr5* promoter polymorphism −2459 A/G (*Hladik et al., 2005*; *Joshi et al., 2017*), we analyzed the

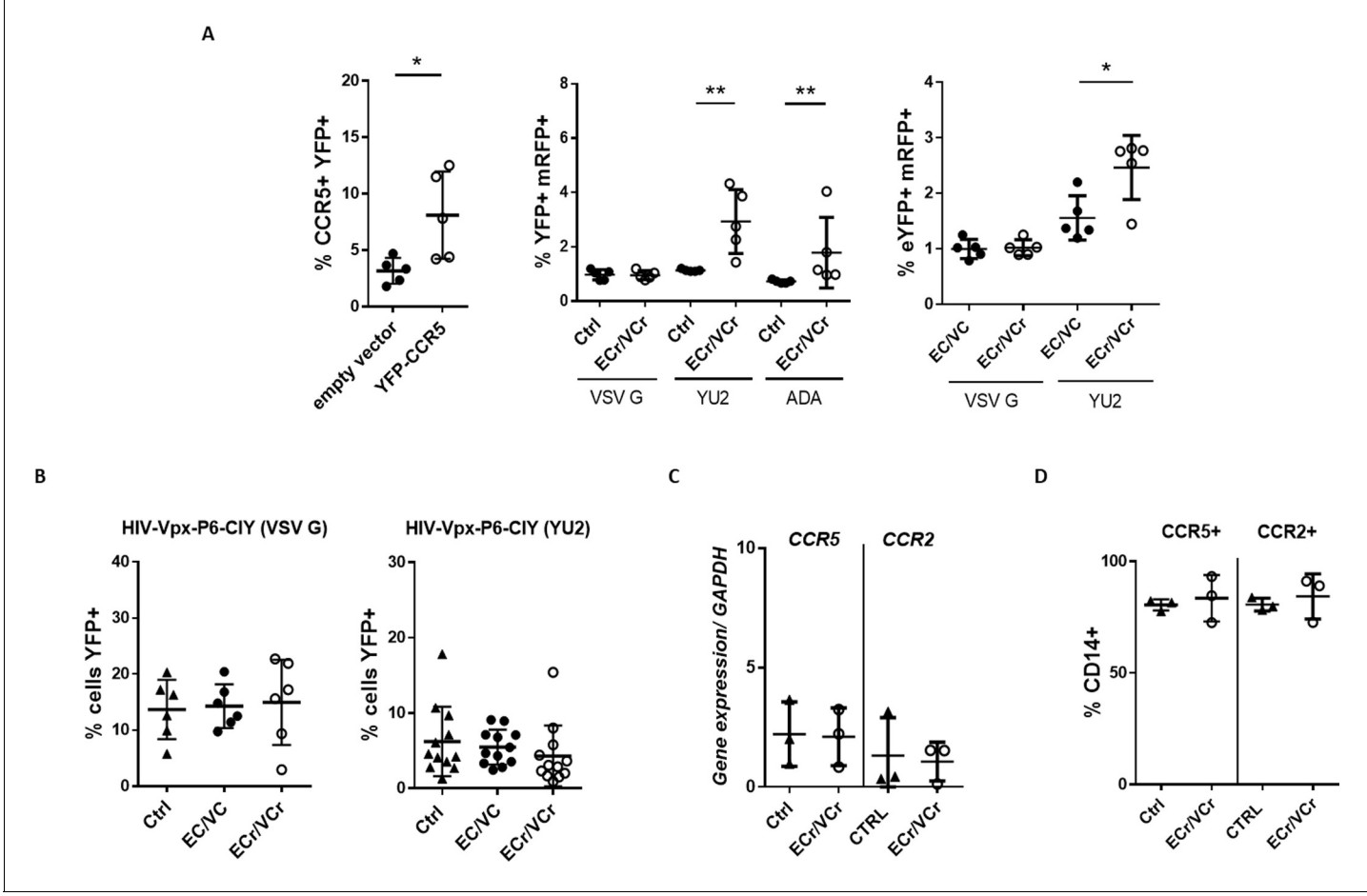

**Figure 4.** Resistance to R5-tropic viruses is due to down-regulation of CCR5 in ECr/VCr. (A) Overexpression of CCR5 in CD4 +T cells using a lentiviral vector (YFP-CCR5). Increased susceptibility to R5-tropic virus after overexpression of CCR5 in EC/VCs with R5 resistance, as measured by YFP+/ mRFP +double positive cells (n = 5 per group). (B) Comparable susceptibility to infection specific to R5-tropic virus in MDMs from ECr/VCr, EC/VCs and Ctrl (n = 6 per group; samples tested in duplicate for YU2). (C–D) Similar *ccr2/ccr5* mRNA (C) and cell surface protein levels (D) in MDMs from EC/VCs (n = 3 per group). Shown in all cases are individual values with Means ± SD, analyzed using U-Mann Whitney test. *p<0.05; **p<0.01.
DOI: https://doi.org/10.7554/eLife.44360.010

frequency of those genotypes in our populations. 76.5% of the Ctrls were A/G heterozygotes, with absence of the polymorphism in 23.5% of the Ctrl population. Interestingly, we only found A/G homozygotes in EC/VC population (8.51%). When analyzed as presence vs. absence of the polymorphism, we identified a lower frequency of homo +heterozygotes in EC/VCs (60.64%) compared to Ctrl (p=0.03). Although a significantly lower frequency was also observed in ECr/VCr (52.38%; p=0.04) compared to Ctrl, we did not observe a significant difference between ECr/VCr and remaining EC/VC (p=0.41). Thus, the presence of this known promoter polymorphism does not contribute to the R5 resistance phenotype in the ECr/VCr population.

By PCR and agarose gel electrophoresis we also analyzed the frequencies of Δ32*CCR5* in our cohort (*Samson et al., 1996*; *Rappaport et al., 1997*), with 14.8% of the Ctrl (4 of 27) being Δ32*CCR5* heterozygotes. We did observe a higher frequency of Δ32*CCR5* heterozygotes in ECr/VCr (33.33% or 7/21) compared to remaining EC/VCs (18.42%, 14/76; p=0.027), suggesting that the presence of this variant contributes in part to the R5 resistance phenotype observed in ECr/VCr subset.

## ATAC-Seq identifies open chromatin regions in ECr/VCr

Given the reduced *ccr2/ccr5* RNA levels observed in ECr/VCr, we decided to examine whether there were differences in chromatin accessibility in this region of chromosome 3, inclusive of *ccr2* and *ccr5*

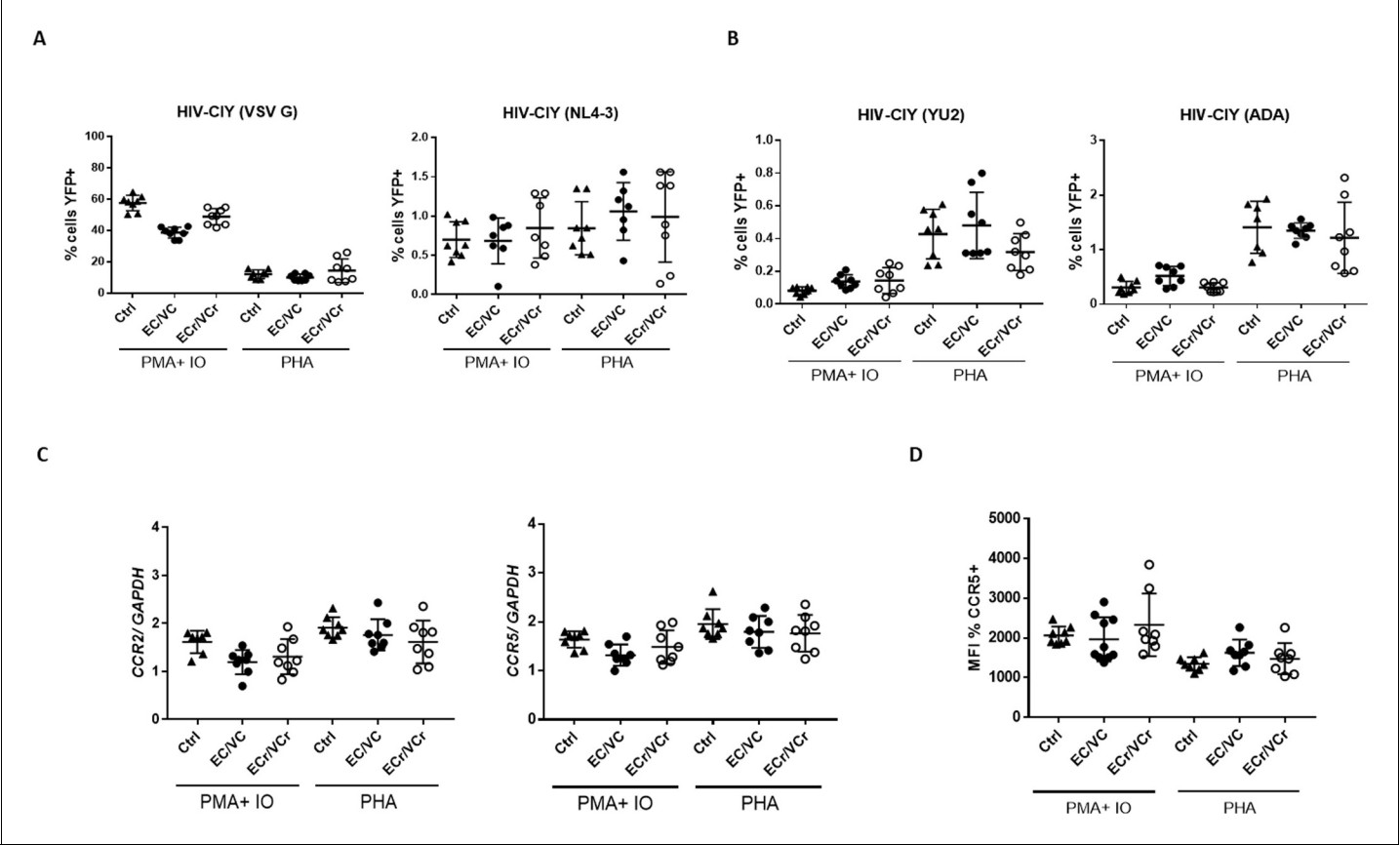

**Figure 5.** Resistance specific to R5-tropic virus is dependent upon T cell activation method. (**A**) Comparable CD4 +T cell susceptibility to X4- and VSV G or (**B**) R5- pseudotyped particles in all groups after PMA plus ionomycin or PHA stimulation. Decreased susceptibility to R5-tropic infection in ECr/VCr compared to Ctrl and remaining EC/VCs after PHA stimulation was not significant. Shown are Means ± SD. (**C**) Comparable *ccr2* and *ccr5* mRNA expression levels between experimental groups after PMA plus ionomycin or PHA treatment. (**D**) Comparable frequency of CCR5 +cells between samples after PMA plus ionomycin or PHA stimulation in activated cells, analyzed as the MFI (n = 8 per experimental group).
DOI: https://doi.org/10.7554/eLife.44360.011

(chr3:45,920,704–46,497,303). DNA libraries were prepared in activated CD4 +T cells from ECr/VCr (n = 4 replicates) and compared to Ctrl samples (n = 4 replicates) and Assay for Transposase Accessible Chromatin with high-throughput sequencing (ATAC-Seq) was performed to quantify differences in open chromatin. Our results identified 64 peaks enriched in ECr/VCr compared to Ctrl (*Figure 6A*), consistent with ~500 kb of highly accessible chromatin in this region of 3p21 in ECr/VCr patients. We explored a small region including *ccr2* and *ccr5* (chr3:46,392,331–46,418,348), and we identified more open chromatin in the *ccr2*- and *ccr5*-promoter regions in ECr/VCr compared to Ctrl (*Figure 6A*).

We also examined chromatin accessibility both upstream and downstream of this ~500 kb region. The coverage matrices of clusters 1 and 2 (upstream) showed a slight increase in ECr/VCr compared to Ctrl whereas there were no observable differences in the downstream ATAC-Seq peaks (*Figure 6—figure supplement 1*). These results suggest that the increase in chromatin accessibility is relatively specific to the ~500 kb region encompassing *ccr2* and *ccr5* in ECr/VCr.

In order to confirm the increased chromatin accessibility in ECr/VCr, we analyzed by ChIP *ccr2* and *ccr5* DNA levels using Tri-Methyl Histone H3 (Lys4) antibody (H3K4Me3) and qPCR (*Figure 6B*). We saw a trend towards greater H3K4Me3 levels in ECr/VCr compared to EC/VC and Ctrl, although differences were not significant (*ccr5: p*=0.42 and p=0.12, respectively). These data, taken together, demonstrate that the down-regulation of *ccr2/ccr5* mRNA levels is accompanied by an increase in open chromatin in ECr/VCr in a specific region of 3p21.

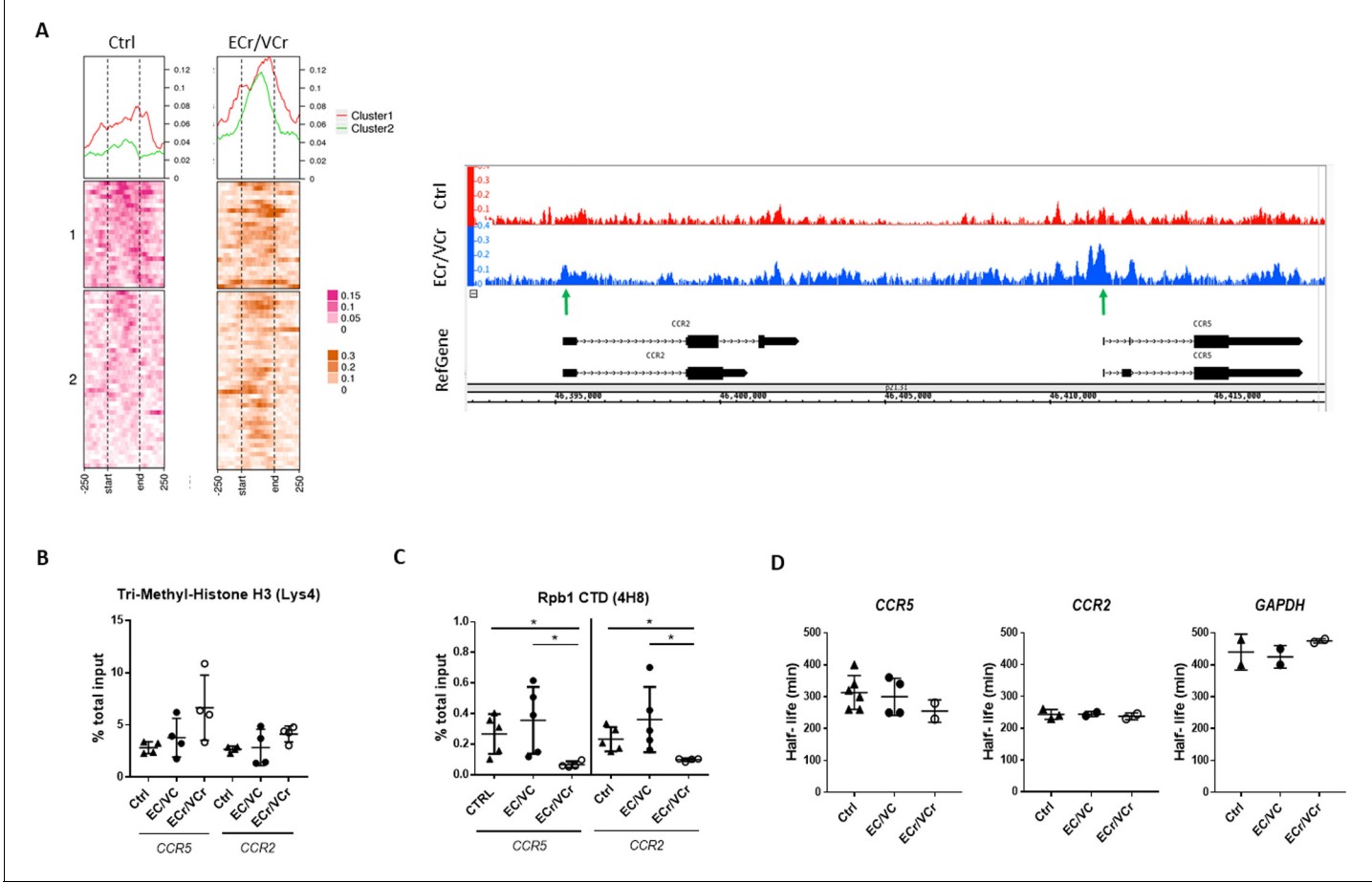

**Figure 6.** Increased chromatin accessibility and lower active transcription in activated CD4 +T cells from ECr/VCr. (**A**) Left panel: ATAC-Seq coverage profiles of region of chr 3p21 (45,920,704-46,497,303) of ECr/VCr CD4 +T cells, compared to those of Ctrl (n = 4 replicates per group). Heat map showing gene TSS aligned, with a window of −250 bp to +250 bp, calculated as a normalized coverage around each TSS. Matrix was divided it into two clusters, based upon Ctrl data. At top is average coverage profile for each of the clusters (cluster one in red and cluster two in green). Right panel: ATAC-Seq peaks of chr 3p21 (46,392,331-46,418,348) of ECr/VCr vs Ctrl visualized using Integrated Genome Browser (IGB), see also *Figure 6—figure supplement 1*. Green arrows highlight increased peaks near the TSS of both genes, *ccr2/ccr5*, in ECr/VCr relative to Ctrl. (**B–C**) ChIP-qPCR, using either Tri-Methyl-Histone H3 (Lys4) (**B**) or Rpb1 (**C**) antibodies, with *ccr2* and *ccr5* DNA quantified by qPCR. Data normalized by the % total input DNA. Shown are Means ± SD (n = 4 and n = 5 per group in B and C, respectively), with statistical analysis performed using Kruskal-Wallis with Dunn's multiple-comparison test. *p<0.05. (**D**) Quantitation of mRNA half-lives of indicated genes in activated CD4 +T cells, using Act D as a transcription inhibitor. T cells were incubated with Act D and harvested (from time 0 to 8 hr). RNA was extracted, and RNA levels quantified by RT-qPCR and half-life calculated using GraphPad PRISM software.

DOI: https://doi.org/10.7554/eLife.44360.012

The following figure supplement is available for figure 6:

**Figure supplement 1.** ATAC-Seq identifies comparable peaks between samples in other regions around chr 3p21.

DOI: https://doi.org/10.7554/eLife.44360.013

### *CCR5* transcriptional down-regulation in ECr/VCr

To determine whether the down-regulation of *ccr2/ccr5* RNA in ECr/VCr was attributable to a decrease in active transcription, we performed ChIP in activated CD4 +T cells using antibodies against Rpb1 CTD, the carboxy terminal domain of the large subunit of RNA polymerase II, followed by qPCR. We observed lower *ccr5* DNA levels in chromatin samples from ECr/VCr (0.069 ± 0.02) compared to those of EC/VCs without the resistance phenotype (0.36 ± 0.21; p=0.02) and Ctrl (0.27 ± 0.13; p=0.03; *Figure 6C*). We also observed comparable results with *ccr2*, with decreased DNA levels in chromatin samples from ECr/VCr (0.09 ± 0.01) compared to remaining EC/VCs

(0.36 ± 0.21; p=0.04) and Ctrl (0.22 ± 0.07; p=0.02). These data are consistent with reduced transcriptional initiation or activity of *ccr2/ccr5* in ECr/VCrs compared to remaining EC/VCs and Ctrl.

We next determined whether the differences in *ccr2/ccr*5 RNA levels were a result of changes in RNA stability. Activated CD4 +T cells were incubated in presence of Actinomycin D for varying lengths of time, RNA isolated, RT-qPCR performed, and RNA half-life calculated from the decay curves for ECr/VCr, Ctrl, and remaining EC/VC populations. We observed comparable half-lives of *ccr2*, *ccr5*, and *gapdh* RNAs in CD4 +T cells from ECr/VCr, Ctrl, and remaining EC/VC groups (*Figure 6D*), indicating that the down-regulation of *ccr2/ccr5* RNA in ECr/VCr was likely a result of differences in transcriptional initiation, rather than due to changes in RNA stability, consistent with the Rpb1 ChIP results above.

## Down-regulation of *ccr2/ccr5* RNA levels in family members of an index VC with R5-tropic resistance

To determine whether there is a hereditary basis associated with R5 resistance, we recruited family members of an index VCr and investigated whether the associated CD4 +T cells had the same in vitro phenotype. Activated CD4 +T cells from several ATL2 family members were infected with pseudotyped viral particles of varying tropisms, and viral susceptibility analyzed by flow cytometry. We observed resistance specific to R5-tropic virus in the T cells of two of three ATL2 family members analyzed, with full susceptibility to X4- and VSV G-pseudotyped HIV (*Figure 7A and B*).

This included the mother and daughter, but not the son. Other family members were not available for testing. Of note, by self-report all family members were HIV seronegative and we were not allowed to do further testing. The percentage of infected cells using R5-tropic virus was significantly lower in activated CD4 +T cells of ATL2 and some family members (ATL2 +FMr 0.51 ± 0.24) compared to those of Ctrl (1.58 ± 0.66; p=0.0015), EC/VC (1.42 ± 0.72; p=0.007), and the family member without the phenotype, FMnr (1.05 ± 0.1; p=0.04).

We next asked whether the observed phenotype seen in family members was associated with down-regulation of *ccr2* and *ccr5* RNA and other genes. RNA-Seq data identified 315 genes significantly differentially expressed between ATL-2 and ATL-2 FMr, compared to Ctrl and ATL-2 FMnr. A complete list of the genes is included in *Supplementary file 2*. More than half (51%, 160/315) were significantly down-regulated in activated CD4 +T cells from ATL2 and FMr compared to Ctrl and FMnr, including *ccr2* and *ccr5* and several genes in 3p21. RT-qPCR confirmed down-regulation in *ccr2/ccr5* RNA levels in activated CD4 +T cells from ATL2 and FM with R5-resistance phenotype (*ccr2*: FMnr 0.24 ± 0.01 vs ATL2 +FMr 0.05 ± 0.02; p=0.03 and *ccr5*: FMnr 0.17 ± 0.005 vs ATL2 +FMr 0.03 ± 0.01; p=0.05, *Figure 7C*). By flow cytometry, we also measured CCR2 and CCR5 cell surface expression in non-stimulated and stimulated CD4 +T cells in EC/VC family members with and without the R5 resistance phenotype (*Figure 7D*). The expression of CCR5 in activated CD4 +T cells from those family members with the resistant phenotype was significantly reduced (% CCR5 +FMnr 12.33 ± 1.55 vs FMr 7.48 ± 2.57; p=0.02). These data point towards a hereditary basis of R5-tropic resistance, at least for the ATL2 pedigree, and that the observed *CCR2/CCR5* down-regulation is genetic in nature.

## Discussion

Here we studied CD4 +T cells purified from PBMCs of 131 EC/VCs and identified a subset of HIV EC/VCs whose T cells were relatively resistant to infection by R5-tropic pseudotyped viral particles, in single cycle, cell-based in vitro assays. This R5-resistance phenotype was associated with transcriptional down-regulation of both *ccr2* and *ccr5*. This same phenotype was observed in family members of an index VC with R5 resistance, and it was also associated with *ccr5* RNA and protein down-regulation, providing strong evidence for a hereditary basis of the phenotype.

The in vitro R5 resistance phenotype was most strongly observed after CD4 +T cell co-stimulation. In agreement with our results, prior studies have demonstrated that PHA-activated CD4 +T cells from ECs were susceptible to both R5- and X4-tropic HIV infection (*Blankson et al., 2007*; *Bailey et al., 2006*; *Sáez-Cirión et al., 2010*). Other groups have demonstrated that anti-CD3-activated CD4 +T cells from ECs were resistant to HIV infection, independent of co-receptor usage (*Chen et al., 2011*; *Sáez-Cirión et al., 2011*; *Paxton et al., 1996*; *Saha et al., 1998*; *Yu and Lichterfeld, 2011*). Only one prior report, from our group, observed T cell resistance specific to R5-tropic

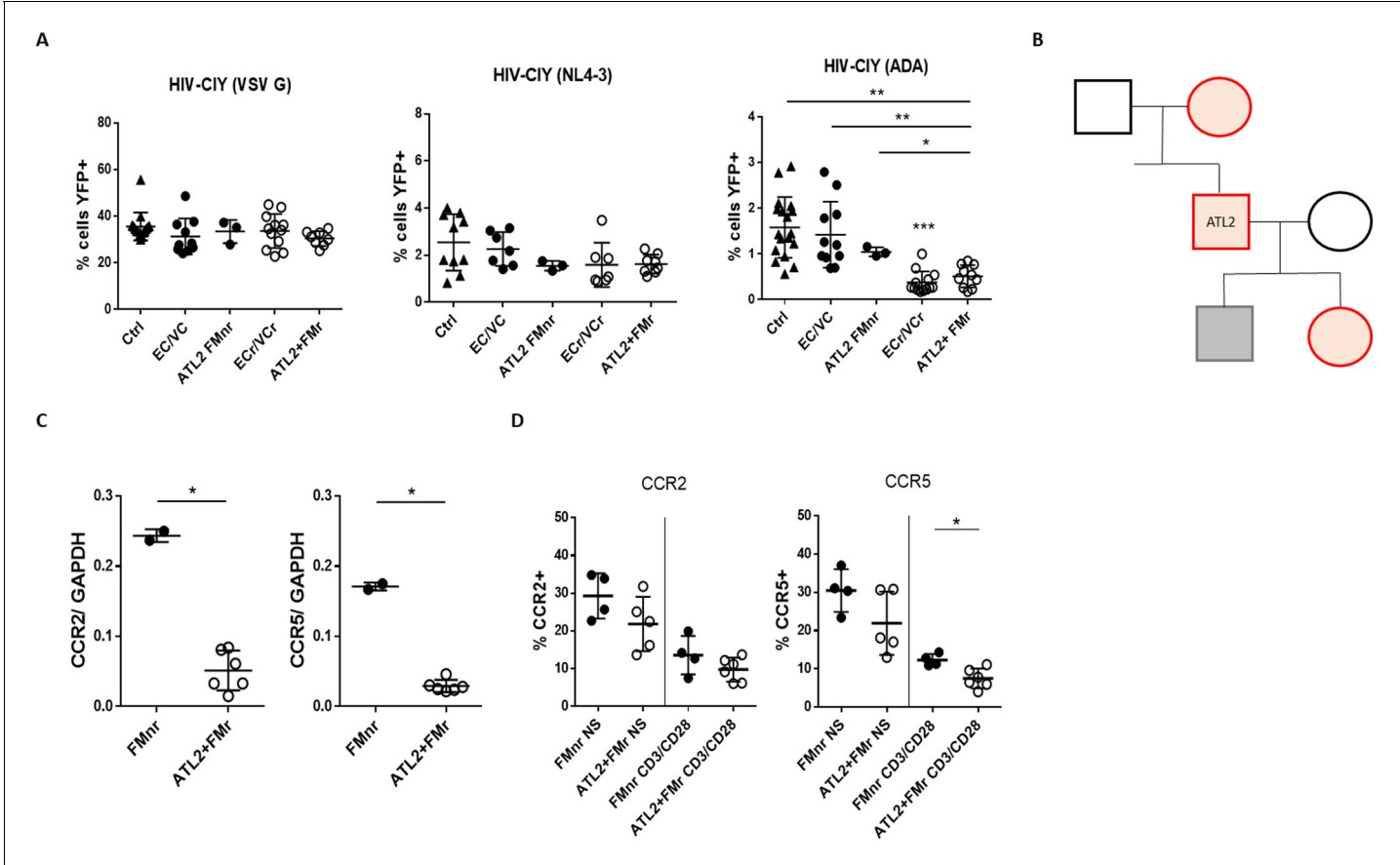

**Figure 7.** Pedigree analysis of an Index VC with R5 resistance phenotype. (**A**) Resistance specific to R5-tropic virus, with equivalent susceptibility to X4- and VSV G, in activated CD4 +T cells from 2 of 3 analyzed ATL2 VC family members. Shown are pooled results from different experiments, with samples tested at least in triplicate. Statistical differences between ECr/VCr and other groups (Ctrl, EC/VC, and ATL2 FMnr) are also shown (**). (**B**) Pedigree analysis of ATL2 EC. Red are individuals with the R5 resistance phenotype (ATL2 FMr); grey represents full susceptibility to infection (ATL2 FMnr); black not available for testing. (**C**) Decreased *ccr2/ccr5* RNA levels in activated CD4 +T cells from family members with R5 resistance. Samples were tested in duplicate. (**D**) Decreased CCR2 and CCR5 surface expression in resting (NS) and activated CD4 +T cells in family members with the resistance phenotype. Samples were tested at least in duplicate; shown are individual values with Mean ±SD. Statistical analysis was performed by using the U-Mann Whitney test or Kruskal-Wallis with Dunn's multiple-comparison test. *p<0.05; **p<0.01. FMr: family member with R5 resistance. FMnr: family member without R5 resistance.

DOI: https://doi.org/10.7554/eLife.44360.014

virus (*Walker et al., 2015*), and the current results are consistent with those data. Our prior study, however, suggested the mechanism was mediated by increased chemokine produced and secreted by activated CD4 +T cells, which would then confer resistance by sterically interfering with Env binding to co-receptor (*Saha et al., 1998*).

In the experiments here, performed on a much larger scale compared to our initial report, chemokine RNA and protein levels were actually decreased in CD4 +T cells of EC/VCs with the R5 resistance phenotype, suggesting that another mechanism was operational. It should be pointed out that there was some overlap in subjects between the two studies. Despite repeated testing, we did not confirm increased chemokine expression in EC11, but instead down-regulation of both *ccr2* and *ccr5* RNA. Of note, this sample was obtained at a later time point, perhaps explaining the observed differences. We also included VCs in the current report, and they were excluded from the previous study. Prior investigations have suggested that CD4 +T cells from ECs retain the ability to proliferate and produce IL-2 in response to HIV (*Emu et al., 2005*) and are highly activated (*Bello et al., 2009*). EC/VCs with the R5-tropic resistant phenotype expressed significantly lower levels of the early activation marker CD69. There were no differences, however, in levels of the late activation marker CD25, which is when the T cells were infected. In addition, those T cells remained fully susceptible

to VSV G- and X4 Env-pseudotyped HIV, thus the significance of the subtly lower CD69 levels in the T cells of the ECr/VCr subset is not known.

Of interest was the fact that we observed the R5 resistance phenotype only in activated CD4 +T cells and not MDMs. The observed phenotype correlates with *ccr2/ccr5* RNA down-regulation in CD4 +T cells, whereas in MDMs there was no down-regulation of those two co-receptor genes, demonstrating a strong correlation between resistance to R5 tropic viruses and down-regulation of *ccr2/ccr5*. It is known that there are large differences in the transcriptional profiles between T cells and MDMs (*Woelk et al., 2004*; *Xue et al., 2014*), and even in T cells different activation protocols result in altered gene expression patterns (*Marrack et al., 2000*; *Xu et al., 2013*). Thus, it is quite conceivable that non-specific T cell stimulation leads to production of transcription factors not present after co-stimulation, resulting in altered RNA and cell-surface levels of CCR5.

The presence of the homozygous *CCR5Δ32* mutation confers protection against mucosal HIV infection (*Liu et al., 1996*; *Samson et al., 1996*), and heterozygotes have slower disease progression (*Rappaport et al., 1997*; *Rodés et al., 2004*). That the frequency of *CCR5Δ32* $\pm$ was significantly higher in EC/VCs with the R5-resistance phenotype compared to other ECs suggests heterozygosity could contribute in part to the R5 resistance phenotype, likely by inactivating one *ccr5* allele and decreasing cell surface expression. Here, we also observed that both *ccr2* and *CCR5* mRNA and cell surface protein levels were down-regulated in ECr/VCrs, supporting the idea that the R5 resistance phenotype is mediated by a transcriptional mechanism. It is unlikely that *CCR5Δ32* affects mRNA levels since nonsense-mediated decay of RNA is not operational if the stop codon is present in the last exon, as it is here. In addition, several lines of evidence presented here favor a transcriptional mechanism for the RNA down-regulation of *ccr2/ccr5*. There was no difference in the half-lives of these RNAs in activated T cells, and ChIP-qPCR data using anti-Rpb1 demonstrated decreased levels of active transcription on *ccr2/ccr5* in ECr/VCrs. Rpb1 is the largest subunit of RNA polymerase II and its presence on DNA correlates strongly with active transcription (*Shin et al., 2016*; *Brookes and Pombo, 2009*; *Phatnani and Greenleaf, 2006*).

Two decades ago, the cis- and trans-acting sequences and factors influencing *ccr5* transcription were studied, and a promoter upstream of *ccr5* was localized and dissected by functional assays (*Liu et al., 1998*; *Mummidi et al., 1997*). Given the lack of upstream sequence conservation and distance of >10 kb, it is highly unlikely that those DNA sequences and transcription factors would also modulate *ccr2* expression. In addition, we observed decreased RNA levels of multiple genes spanning ~500 kb of 3p21, both centromeric and telomeric to *ccr2/ccr5*, consistent with a more global and coordinate down-regulation of multiple chemokines and their receptors in the activated CD4 +T cells from ECr/VCr.

ATAC-Seq is an established method for quantifying chromatin accessibility in different cell populations (*Corces et al., 2016*). Previous reports have suggested that histone modifications upstream of coding regions play a role in transcriptional regulation (*Bernstein et al., 2002*). In general, H3K4Me3 is associated with open chromatin, specifically marking the promoters of active genes, and correlates with higher levels of transcripts (*Heintzman et al., 2007*; *Bernstein et al., 2005*). In our study, however, we observe CD4 +T cells from ECr/VCr have more open chromatin over ~500 Kb region in chr3, including *ccr2* and *ccr5*, which surprisingly was associated with lower transcription of both genes. It had also been shown that levels of DNA methylation in the *ccr5* locus correlated inversely with CCR5 levels on T cells (*Gornalusse et al., 2015*), which is also a typical transcriptional control mechanism. The fact that CD4 +T cells of ECr/VCr have decreased transcriptional initiation/ transcript levels of *ccr2/ccr5* and yet more open chromatin suggests that there is a dissociation between chromatin access and transcription of these genes, for inapparent reasons.

Interestingly, two of the three family members of an Index VCr had CD4 +T cells with a similar R5 resistance phenotype, with associated down regulation of CCR5 RNA and protein levels. The fact that it was multi-generational and in both sexes is highly suggestive but is not definitive evidence that the phenotype is autosomal dominant. Additional family studies will be necessary to determine whether the R5 tropic resistance phenotype has hereditary dominance. Autosomal dominant inheritance would be consistent with altered cell signaling or DNA binding factor, acting in a trans-dominant fashion and negatively influencing transcription of both *ccr2/ccr5* alleles (*Liu et al., 1998*). Precedents include naturally-occurring dominantly suppressive variants of human *stat5* (*Crotti et al., 2007*; *Yamashita et al., 2003*), those of human *stat6* that are amino terminus truncated for the SH2 domain (*Mikita et al., 1996*; *Patel et al., 1998*), or an alternatively spliced form of human *stat3* that

functioned as a dominant negative regulator of transcription (*Zammarchi et al., 2011*). The JAK/STAT signaling pathway is important for expression of multiple chemokines and their receptors, including *ccr5*, and becomes activated after T cell co-stimulation (*Shuai and Liu, 2003*; *Wong and Fish, 1998*; *Zi et al., 2017*). It is an open question whether T cell co-stimulation leads to the production of a dominant-negative transcription factor in the ECr/VCr subset, resulting in reduced *ccr2/ccr5* or more global transcriptional down-regulation.

LOC102724291 is transcribed antisense to *ccr5* and it has been suggested that *loc102724291* may contribute to virus set-point (*McLaren et al., 2015*). Our results revealed a down-regulation in *loc102724291* RNA levels in activated T cells in ECr/VCr, with no correlation between *ccr2* and *ccr5* gene expression, making it unlikely that it is modulating the expression of those genes. Without invoking a more global mechanism of transcriptional control, it is difficult to understand how *loc102724291* would be capable of inhibiting transcription of other genes in that chromosome region. Instead, it appears that lncRNA may be similarly down-regulated to other genes in the region.

In conclusion, our data suggest that the R5-tropic resistance phenotype seen in a subset of EC/VCs is associated with transcriptional down-regulation of *ccr5*, which appears to be heritable, across multiple generations. That the chromatin of this region of 3p21 appears to be more accessible yet multiple genes are down-regulated implies a complex but coordinate mode of transcriptional regulation. Because these ECs are able to persistently suppress viral replication, further investigation into the mechanisms underlying these findings should inform the HIV cure effort.

# Materials and methods

**Key resources table**

| Reagent type (species) or resource | Designation | Source or reference | Identifiers | Additional information |
|---|---|---|---|---|
| Antibody | anti-CD3 mouse Monoclonal Antibody (OKT3), PerCP-Cyanine5.5 | eBioscience | Cat # 45-0037-42; RRID: AB_10548513 | Dilution (1:100) |
| Antibody | anti-CD4 mouse Monoclonal Antibody (RPA-T4), APC | eBioscience | Cat # 17-0049-42; RRID: AB_1272048 | Dilution (1:100) |
| Antibody | anti-CD14 mouse Monoclonal Antibody (61D3), FITC | eBioscience | Cat # 11-0149-42; RRID: AB_10597597 | Dilution (1:100) |
| Antibody | anti-CD8a mouse Monoclonal Antibody (HIT8a), PE | eBioscience | Cat # 12-0089-42; RRID: AB_10804039 | Dilution (1:100) |
| Antibody | CD3 mouse Monoclonal Antibody (OKT3), Functional Grade | eBioscience | Cat # 16-0037-81; RRID: AB_468854 | 10 µg/ml |
| Antibody | CD28 mouse Monoclonal Antibody (CD28.2), Functional Grade | eBioscience | Cat # 16-0289-81; RRID: AB_468926 | 4 µg/ml |
| Antibody | CD25 mouse Monoclonal Antibody (BC96), PE | eBioscience | Cat # 12-0259-42; RRID: AB_1659682 | Dilution (1:200) |

*Continued on next page*

*Continued*

| Reagent type (species) or resource | Designation | Source or reference | Identifiers | Additional information |
|---|---|---|---|---|
| Antibody | CD69 mouse Monoclonal Antibody (FN50), FITC | eBioscience | Cat # 11-0699-42; RRID: AB_10853975 | Dilution (1:200) |
| Antibody | CD45RA mouse Monoclonal Antibody (HI100), FITC | eBioscience | Cat # 11-0458-42; RRID: AB_11219672 | Dilution (1:100) |
| Antibody | CD45RO, mouse Monoclonal PE-Cyanine5, clone: UCHL1 | eBioscience | Cat # 15597726; Gene ID: 5788 | Dilution (1:100) |
| Antibody | PE anti-human CD195 (CCR5) rat Monoclonal Antibody | Biolegend | Cat # 313707; RRID: AB_345307 | Dilution (1:100) |
| Antibody | APC anti-human CD192 (CCR2) mouse Monoclonal Antibody | Biolegend | Cat # 357207; AB_2562238 | Dilution (1:100) |
| Antibody | anti-Rpb1 CTD mouse Monoclonal | Cell Signaling | Cat # 2629; 4H8 | ChIP (1:50) |
| Antibody | Tri-Methyl-Histone H3-Lysine 4 (H3Lys4) rabbit Monoclonal | Cell Signaling | Cat # 9727 | ChIP (1:50) |
| Peptide, recombinant protein | Recombinant Human IL-2 | *E. coli*-derived human IL-2 protein | R and D: P60568 | |
| Recombinant DNA reagent | HIV-cycT1-IRES-YFP (HIV-CIY) | this paper | Sutton lab | plasmid |
| Recombinant DNA reagent | pSM-ADA Env | this paper | Sutton lab | plasmid |
| Recombinant DNA reagent | pSRα-YU2 Env | this paper | Heinrich Gottlinger, UMass Medical Cener | plasmid |
| Recombinant DNA reagent | pSRα-NL4-3 Env | this paper | Heinrich Gottlinger, UMass Medical Cener | plasmid |
| Recombinant DNA reagent | pME-VSV G | this paper | Sutton lab | plasmid |
| Recombinant DNA reagent | pCCL3L1 | Origene | NM_021006.4, NP_066286 | plasmid |
| Recombinant DNA reagent | pCCL4 | this paper | generated by PCR using pcDNA3/1 + CAT plasmid; Sutton lab | plasmid |
| Recombinant DNA reagent | Vpx-myc-his | | Ned Landau laboratory, NYU Medical Center | plasmid |
| Recombinant DNA reagent | pMDL-Chp6 | | Ned Landau laboratory, NYU Medical Center | plasmid |
| Cell line (H. Sapiens) | HEK 293T | ATCC | Cat# CRL-3216, RRID:CVCL_0063 | |
| Cell line (H. Sapiens) | GHOST.Hi5 | NIH AIDS Reagent Program | NIH-ARP Cat# 3944–343, RRID:CVCL_1E17 | |

*Continued on next page*

*Continued*

| Reagent type (species) or resource | Designation | Source or reference | Identifiers | Additional information |
|---|---|---|---|---|
| Cell line (H. Sapiens) | GHOST.CXCR4 | NIH AIDS Reagent Program | NIH-ARP Cat# 3685–448, RRID:CVCL_S492 | |
| Cell line (H. Sapiens) | TZM-bl cells | NIH AIDS Reagent Program | NIH-ARP Cat# 8129–442, RRID:CVCL_B478 | |
| Commercial assay or kit | RNeasy Mini Kit | Qiagen | ID: 74104 | |
| Commercial assay or kit | Mouse MIP-1 alpha (CCL3) ELISA | Invitrogen | LS885601322 | |
| Commercial assay or kit | Human CCL4 (MIP-1 beta) ELISA | Invitrogen | Invitrogen 88703476 | |
| Commercial assay or kit | High-Capacity cDNA Reverse Transcription Ki | ThermoFisher | ID: 4368814 | |
| Commercial assay or kit | DNeasy blood and tissue kit | Qiagen | Cat No./ID: 69504 | |
| Commercial assay or kit | SimpleChIP enzymatic ChIP kit agarose beads | Cell Signaling | Cat #9002 | |
| Commercial assay or kit | MinElute Reaction Cleanup kit | Qiagen | Cat No./ID: 28204 | |
| Commercial assay or kit | Transposase mixture | Illumina | Nextera DNA library prep kit; FC-131–1024 | |
| Chemical compound, drug | Phorbol 12-myristate 13-acetate | Sigma | PubChem CID: 27924 | |
| Chemical compound, drug | Ionomycin calcium salt | Sigma | I3909 | |
| Chemical compound, drug | Actinomycin D | Sigma. From Streptomyces sp | Cat # A1410 | |
| Chemical compound, drug | Digitonin | Promega | G944A | |
| Other | Power SYBR Green PCR Master Mix | ThermoFisher | Cat # 4367659 | Commercial reagent |
| Other | NEBnext PCR master mix | New England BioLabs | Cat # M0541S | Commercial reagent |
| Software, algorithm | CummeRbund | R package version 2.24.0 | DOI: 10.18129/B9.bioc.cummeRbund | |
| Software, algorithm | Illumina's CASAVA 1.8.2 | Illumina | Ref. 15011197 | |
| Software, algorithm | GraphPad Prism | GraphPad Prism (https://graphpad.com) | RRID:SCR_015807 | |
| Software, algorithm | FlowJo | https://www.flowjo.com/solutions/flowjo | RRID:SCR_008520 | |

## Study subjects

131 HIV +EC/VC subjects were recruited from Yale New Haven Hospital and other HIV clinics in USA. Inclusion criteria for EC/VCs were HIV seropositivity and plasma VL < 50 (ECs) or 50 < VL < 2000 (VCs) for at least 6–12 months in the absence of ART, except in some special circumstances, as specified (*Supplementary file 1*). Occasional viral blips were allowed but not virologic escape or

clear trends in viremia. Exclusion criteria included contraindication to peripheral phlebotomy and inability to provide informed consent. Clinical characteristics recorded included gender, age, CD4 +T cell count, VL, and year of HIV diagnosis. Also, HIV acquisition risk factor, major comorbidities, and protective HLA alleles data were collected, if known. The study was approved by both the Yale IRB (Yale New Haven Hospital and other Yale-affiliated HIV clinics in Connecticut), and the local IRBs (the SCOPE cohort from UCSF, the Ragon Institute of MGH, MIT and Harvard, and from Veterans Medical Center HIV clinics from Atlanta and Dallas) and informed, written consent was obtained from all subjects.

Anonymized, leukocyte-enriched fractions of peripheral blood from 35 normal, healthy donors were obtained and used as controls. Three family members (FM) of an Index VC (Atl2) were enrolled and whole blood obtained by peripheral phlebotomy. Based upon self-report, all FM included in the study were HIV seronegative. CFAR relies on self-reporting with respect to HIV-uninfected cases. Our IRB protocol did not allow us to perform HIV testing on FM because of privacy concerns.

## Peripheral blood mononuclear cell collection and CD4 +T cell purification

Mononuclear cells were obtained after Ficoll-Paque PLUS (GE Healthcare Life Sciences, Piscataway, NJ) centrifugation of leukocyte-enriched fractions of whole blood. CD4 +T cells were purified by positive selection, using anti-CD4 magnetic microbeads (Miltenyi Biotech, San Diego, CA) following the manufacturers' recommendations. The purity of the CD4 +T cells was confirmed by flow cytometric analysis using anti-human CD3-PerCP-Cyanine5.5 (clone OKT3; eBioscience, San Diego, CA) and CD4-APC (clone RPA-T4; eBioscience) antibodies. Antibodies against human CD14 and CD8 were included to confirm absence of contaminating monocytes and CD8 +T cells (anti human CD14-FITC, clone 61D3; anti-human CD8a-PE, clone HIT8a; eBioscience). Purity of CD4 +T cells was >95%. The remaining cells were predominantly CD4-low monocytes with <1% contaminating CD8 +T cells. T cells were resuspended in staining buffer (2% FBS in PBS) on ice for 30 min, washed, and then placed in IC fixation buffer (eBioscience) on ice for 10 min. Cells were washed, resuspended in staining buffer, and analyzed by flow cytometry (LSRII, BD; Franklin Lakes, NJ). Data were analyzed using FlowJo software (version 10.1 Ashland, OR).

## CD4 +T cell activation and staining

CD4 +T cells were activated for 72 hr, using tissue culture plates pre-coated with 1 µg/mL anti-CD3 (clone OKT3; eBioscience) in the presence of 2 µg/mL soluble anti-CD28 (clone 28.2; eBioscience) and 100IU/mL IL-2 (recombinant, R and D Systems, Minneapolis, MN). To check activation status, activated CD4 +T cells were analyzed by light microscopy to confirm refractility and aggregation. The percentage of activated cells was calculated by flow cytometry as above, using anti-human CD25-PE (clone BC96) and CD69-FITC (clone FN50; eBioscience) antibodies. Percentage of naïve and memory CD4 +T cells was analyzed using anti-human CD45RA-FITC (clone HI100) and CD45RO-PeCy5 (clone UCHL1; eBioscience), respectively. To differentiate CM from EM T cells, activated CD4 +T cells were stained with CD45RO-PeCy5 and CD27-FITC (clone M-T271; BD) and analyzed by flow cytometry. To assess CCR2 and CCR5 cell surface levels, non-activated and activated CD4 +T cells were stained for 30 min with fluorescently labeled antibodies against human CD195-PE (CCR5; clone HEK/1/85a; Biolegend, San Diego, CA) or CD195-APC (clone 3A9; BD), and CD192-APC (CCR2; clone K036C2; Biolegend). PE-rat IgG2a, *k* (clone RTK2758) and APC-mouse IgG2a, *k* (clone MOPC-173) antibodies were used as isotype controls (Biolegend). Cells were fixed, resuspended in 2% FBS in PBS, and analyzed by flow cytometry as percentage of positive cells and as MFI.

Alternatively, T cells were activated using 1 mg/ml phytohaemagglutinin (PHA; Sigma-Aldrich, St. Louis, MO), or 10 ng/ml PMA (Sigma) plus 500 ng/ml ionomycin (Sigma) for 72 or 48 hr, respectively, in the presence of 100IU/ml IL-2.

## Cell transfection, virus production and single cycle HIV infection

Pseudotyped lentiviral particles were produced by transient transfection of 293 T cells using the calcium phosphate method and the following plasmids: HIV-cycT1-IRES-YFP (HIV-CIY) as packaging/transfer vector, pSM-ADA Env and pSRα-YU2 Env (both R5-tropic), and pSRα-NL4-3 Env (X4-tropic), with pME-VSV G (pan-tropic control). Viral particles were harvested 72 hr after transfection and

frozen after confirming the efficiency of the transfection by flow cytometry and fluorescence microscope observation. Vector supernatants were tested on GHOST HI5 (R5-tropic) or GHOST CXCR4 (X4-tropic) cells by end-point dilution and also by flow cytometry, with a range of infectivity between $2.5 \times 10^5$ U/ml to $3.0 \times 10^6$ U/ml. VSV G pseudotyped particles were used as positive control, with an infectivity of $\sim 2.5 \times 10^7$ U/ml. For normalization purposes, for each pseudotyped virus the same amount of IU was used to infect activated CD4 +T cells in the same total volume and plate format by spinoculation at 1800 rpm for 30 min, and at 72 hr percentage of YFP+T cells was quantified by flow cytometry.

## HIV replication-competent assay

$1 \times 10^5$ activated CD4 +T cells (anti-CD3/CD28) were infected in triplicate with 0.001 ml of pNL-BaL or 0.01 ml of HIV-NL4-3ΔR1, in the presence of IL-2ample l of PBS and lysis in 201 CD28)U using Luciferase assay.se to infect TzmBL cells . Both of these viruses were prepared by plasmid co-transfection of 293 T cells with pME VSV G, to facilitate initial rounds of viral replication. On alternate days post-infection (from day 1 to 21), supernatant was removed, centrifuged, and used to infect 10,000 TZM-bl cells (obtained from the NIH AIDS Reagent Program). Reporter cells were harvested 72 hr post-infection, washed with 0.5 ml of ost-infection, akes, NJ). PBS and lysed in 0.2 ml of lysis buffer (25 mM Tris-phosphate (pH 7.8), 2 mM DTT, 2 mM 1,2-diaminocyclohexane-N,N,N´,N´-tetraacetic acid, 10% glycerol, and 1% Triton X-100). FFLUC assay was performed by incubating 0.1 ml of lysate with 0.1 ml of assay buffer (25 mM Gly-Gly, 15 mM potassium phosphate pH 7.8, 15 mM magnesium sulfate, 4 mM EGTA, 2 mM ATP and 1 mM DTT) and 0.015 ml Luciferin solution (0.2 mM, Sigma). Bioluminescence was immediately measured in a Gen5 (BioTek) Instrument (Winooski, VT).

## Enzyme-linked immunosorbent assays and conditioned media transfer

CD4 +T cells were activated for 3 days with anti-CD3/CD28 in presence of IL-2 and culture supernatants were harvested and frozen at −80 degrees. Human MIP-1α (CCL3) and MIP-1β (CCL4) instant ELISA kits (eBioscience) were used to measure chemokine levels in culture supernatants, according to the manufacturer's instructions. Media transfer experiments were performed to investigate whether soluble factors were responsible for the inhibition of HIV replication. Activated CD4 +T cells from healthy controls were incubated in presence of supernatant from activated CD4 +T cells from EC/VCs and Ctrl and T cells were then infected with different pseudotyped HIV particles. As control, we included supernatants from 293 T cells transfected with the following plasmids: (i) pCCL3L1 encoding MIP1α (Origene, Rockville, MD); (ii) pCCL4 encoding MIP1β (generated by PCR-amplifying the ccl4 coding sequence from human cDNA and ligating the product into pcDNA3/1 + CAT plasmid). After 30 min of incubation with culture supernatant, cells were infected with pseudotyped HIV particles. T cells were harvested after three days and infectivity was analyzed by flow cytometry for YFP conferred by virus infection.

## RNA-Seq

High quality RNA was isolated from $1 \times 10^6$ activated CD4 +T cells (aCD3/CD28) using the RNeasy Mini kit (Qiagen, Germantown, MD). RNA integrity was verified by running an Agilent Bioanalyzer gel. For the RNAseq library preparation, mRNA was purified from total RNA with oligo-dT beads and sheared by incubation at 94 degrees. Following first-strand synthesis with random primers, second strand synthesis was performed with dUTP for generating strand-specific sequencing libraries. The cDNA library was then end-repaired, and A-tailed, adapters ligated, and second-strand digestion was performed by U-DNA-Glycosylase. Indexed libraries that meet appropriate cut-offs were quantified by qRT-PCR and insert size distribution determined with the LabChip GX or Agilent Bioanalyzer. Samples were sequenced using 75 bp single or paired-end sequencing on an Illumina HiSeq 2500 according to Illumina protocols. Signal intensities were converted to individual base calls during a run using the system's Real Time Analysis software. Multiplexing and alignment to the human genome was performed using Illumina's CASAVA 1.8.2 software. DNA sequence data generated were stored in FASTQ format and quality control was performed using FastQC version 0.10.1. Quality-filtered reads (low quality reads <20 were removed) were aligned to sequences of the human genome (hg19) downloaded from Illumina's iGenome resource (Illumina, San Diego, CA), as previously described (Garber et al., 2011). Reads were analyzed using Cuffdiff (Trapnell et al., 2012) in

order to allow estimation of differential gene expression using functions of the R package 'cummeRbund'.

## Reverse transcription and real time quantitative PCR

RNA levels of *ccr2, ccr5, cxcr4, cd4, ccr1, ccr3, fyco1, cxcr6* and *loc102724297* were measured by real time quantitative PCR (RT-qPCR). Total RNA was extracted from activated CD4 +T cells using the RNeasy mini kit (Qiagen). $A_{260/280}$ was determined to confirm the RNA was of high quality, and 1 µg was used for first-strand complementary DNA synthesis using High Capacity cDNA Transcription Kit (Life Technologies; Warrington, UK). Quantitative RT-PCR was performed on an Applied Biosystems 7500 Fast Real-Time PCR System using Power SYBR Green PCR Master Mix (Life Technologies) and the following primers:

*ccr5*-F:5'-AAAAAGAAGGTCTTCATTACACC-3' and *ccr5*-R:5'-CTGTGCCTCTTCTTCTCATTTCG-3'; *ccr2*-F:5'-CACATCTCGTTCTCGGTTTATC-3' and *ccr2*-R:5'-AGGGAGCACCGTAATCATAATC-3'; *cd4*-F:5'-TGCCTCAGTATGCTGGCTCT-3' and *cd4*-R:5'-GAGACCTTTGCCTCCTTGTTC-3'; *cxcr4*-F:5'-CTACACCGAGGAAATGGGCT-3' and *cxcr4*-R:5'-CCACAATGCCAGTTAAGAAGA-3'; *fyco1*-F:5'-CGCCTCACTTGCTTGGTAG 3' and *fyco1*-R:5'-CTGTGTGGTAGTCCTCCTCC-3'; *cxcr6*-F:5'-GACTATGGGTTCAGCAGTTTCA-3'and*cxcr6*-R:5'-GGCTCTGCAACTTATGGTAGAAG-3'; *ccr1*-F:5'-ACTATGACACGACCACAGAGT-3' and *ccr1*-R:5'-CAACCAGGCCAATGACAAATA-3'; *ccr3*-F:5'-GTCATCATGGCGGTGTTTTTC-3' and *ccr3*-R:5'-CAGTGGGAGTAGGCGATCAC-3'; *loce1*-2 F:5'-CTCACCAGTGTTCGCAGAAA-3' and *loce1*-2 R:5'-TCATGTAGGTGCAGGCAGAC-3'; *loce3*-F:5'-GCATCTCACTGGAGAGGGTTT-3'and*loce3*-R:5'-TTTGCAGAGAGATGAGTCTTAGC-3'; *gapdh*-F:5'-TTGCCATCAATGACCCCTT-3' and *gapdh*-R:5'-CTCCACGACGTACTCAGCG-3'.

For relative quantification, we compared the amount of target to the values obtained for *gapdh* as a normalization control. Data obtained were compared to a standard curve generated by serial dilution of a template complementary DNA and expressed as target *gene:gapdh* ratios.

## Overexpression of CCR5 in activated CD4+ T cells and single-cycle assay

To confirm that the R5-resistance to infection in EC/VC was due to down-regulation of CCR5, we overexpressed CCR5 in EC/VC T cells with R5-tropic resistance in comparison to those of EC/VC without the phenotype and Ctrl, and those T cells were then infected with HIV pseudotyped particles to determine whether they now had increased susceptibility to R5 virus. CD4+ T cells activated by anti-CD3/CD28 co-stimulation were first transduced with VSV G-pseudotyped HIV vector encoding both CCR5 and YFP (pHIV-CCR5-IRES-YFP) or YFP alone (HIV-IRES-YFP). T cells were then infected with an HIV vector encoding mRFP and pseudotyped with either R5 Envelopes or VSV G. After 72 hr, cells were analyzed by flow cytometry to quantify the percentage of double positive cells (YFP+/mRFP+), normalized to HIV-IRES-YFP transduction results.

## *CCR5Δ32* and promoter polymorphism detection by PCR

Genomic DNA extracted from mononuclear cells was purified using DNeasy blood and tissue kit (Qiagen). *CCR5* genotype (Δ32 vs. WT) was determined by agarose gel electrophoresis following PCR using the following primers: *CCR5 Δ32* F:5'-ATAGGTACCTGGCTGTCGTCCAT-3'; *CCR5 Δ32* R:5'-GATAGTCATCTTGGGGCTGGT-3' (*de Roda Husman et al., 1997*). Promoter polymorphism A/G −2459*CCR5* was performed by restriction fragment length polymorphism analysis as previously described (*McDermott et al., 1998*), using the following primers *CCR5* 2459 F:5'-CCGTGAGCCCATAGTTAAAACTC-3'; *CCR5* 2459 R:5'-CACAGGGCTTTTCAACAGTAAGG-3'. PCR products were electrophoresed on a 2% agarose gel and genotypes were determined by visual inspection of ethidium bromide stained banding pattern.

## Measurement of mRNA stability

CD4 +T cells activated by anti-CD3/CD28 co-stimulation were treated with 5 µg/ml Actinomycin D (Sigma) for varying lengths of time. *ccr2, ccr5*, and *gapdh* mRNA levels were quantified at each time point by RT-qPCR using SYBR Green. mRNA decay and half-lives were calculated using a time-point standard curve.

## ATAC-Seq

ATAC-Seq was performed as previously described (*Buenrostro et al., 2015*), with some modifications. CD4 +T cells were activated with anti-CD3/CD28 in presence of IL-2 for 3 days. 50,000 cells were lysed and transpositions were performed using transposase mixture (Nextera DNA library prep kit, Illumina), supplemented with 0.01% digitonin (Promega; Madison, WI). Transposition reactions were incubated for 30 min at 37°C in a ThermoMixer (Eppendorf) with agitation at 300 rpm. DNA was purified using the MinElute Reaction Cleanup kit (Qiagen), and libraries amplified using NEBnext PCR master mix with the following primers:

Ad1_noMX:AATGATACGGCGACCACCGAGATCTACACTCGTCGGCAGCGTCAGATGTG;
Ad2.1_TAAGGCGA:CAAGCAGAAGACGGCATACGAGATTCGCCTTAGTCTCGTGGGC
TCGGAGATGT;
Ad2.2_CGTACTAG:CAAGCAGAAGACGGCATACGAGATCTAGTACGGTCTCGTGGGC
TCGGAGATGT;
Ad2.3_AGGCAGAA:CAAGCAGAAGACGGCATACGAGATTTCTGCCTGTCTCGTGGGC
TCGGAGATGT;
Ad2.4_TCCTGAGC:CAAGCAGAAGACGGCATACGAGATGCTCAGGAGTCTCGTGGGC
TCGGAGATGT;
Ad2.5_GGACTCCT:CAAGCAGAAGACGGCATACGAGATAGGAGTCCGTCTCGTGGGC
TCGGAGATGT;
Ad2.6_TAGGCATG:CAAGCAGAAGACGGCATACGAGATCATGCCTAGTCTCGTGGGC
TCGGAGATGT.

Libraries were quantified using RT-qPCR prior to sequencing. All Fast-ATAC libraries were paired-end sequenced, 75 bp using a HiSeq2500 instrument. Quality of FASTQ files was performed using FASTX trimmer. More than 50 million reads were mapped, with <10% mapped, on average, to the mitochondrial genome. The reads were aligned to the hg19 (UCSC) version using Burrows-Wheeler Aligner (BWA-MEM). Peaks were called using MACS2 (*Zhang et al., 2008*) peak-caller, and the reads from input DNA sample were used as control. Visualization of the peaks was done using R Software.

## Chromatin immunoprecipitation-qPCR

Chromatin immunoprecipitation (ChIP) was performed using SimpleChIP enzymatic ChIP kit agarose beads (Cell Signaling) according to the manufacturer's protocol. Three million CD4 +T cells were activated for 3 days with anti-CD3/CD28. Cells were fixed, and chromatin was sonicated after digestion with micrococcal nuclease. IP was performed with anti-Rpb1 CTD (4H8; Cell Signaling, #2629) or anti-Tri-Methyl-Histone H3-Lysine 4 (H3Lys4) mouse monoclonal antibody (Cell Signaling, #9727), with Histone H3 XP and rabbit IgG serving as positive and negative controls, respectively. DNA was purified by spin column, measured, and amplified by RT-qPCR to quantify *ccr2* and *ccr5* DNA. Primers for *gapdh* were used as a control.

## Generation of human monocyte-derived macrophages and infectivity assays

Mononuclear cells were obtained via peripheral phlebotomy and Ficoll-Paque density gradient centrifugation. Monocytes were purified using anti-human CD14 +microbeads (Miltenyi). Cell purity was confirmed by flow cytometry using anti-CD14-FITC antibody (eBoscience). To differentiate monocytes to macrophages, monocytes were cultured for 7 days in RPMI 1640 supplemented with 10% FBS and 10 ng/ml M-CSF (eBioscience), adding fresh growth factor every 2 days. CCR2 and CCR5 cell surface expression was assessed by FACS analysis. Macrophages were then infected using HIV-CIY prepared with Vpx-myc-his and pMDL-Chp6 (kind gifts of Ned Landau, NYU Medical Center), pseudotyped with either R5 Envelope or VSV G. Macrophages were analyzed by flow cytometry after 72 hr to determine infection efficiency.

## Cell lines

HEK 293 T cells were originally obtained from ATCC and authenticated by transfection testing in vitro, their gross morphology, resistance to 1 mg/ml G418, susceptibility to first generation adenoviral vectors, and growth characteristics. GHOST.Hi5 and GHOST.CXCR4 cells were obtained from the NIH AIDS Reagent Program. Their identity was authenticated by gross morphology, growth

characteristics, expression of eGFP after infection with HIV of the appropriate tropism, confirmation of CCR5 (GHOST.Hi5) and CXCR4 (GHOST.CXCR4) cell surface expression by flow cytometry, and also testing for CD4 expression (both lines). TZM-bl cells were also obtained from the NIH AIDS Reagent Program and authenticated by gross morphology and growth characteristics, cell surface expression of both co-receptors and CD4 by flow cytometry, and susceptibility in vitro to HIV, with readout being both FFLUC activity in infected cell lysates and lacZ expression in fixed cells, the latter using X-Gal. All cell lines were tested to confirm absence of mycoplasma contamination.

## Statistics

Correlations between mRNA and cell surface expression levels, and percentage of infected CD4 +T cells were assessed by Spearman´s test. Statistical differences between groups were determined using Mann-Whitney U test for two independent samples or one-way ANOVA using Kruskal-Wallis non-parametric test, as required. Frequencies of HLA alleles and presence of polymorphisms were compared between groups using Chi-Square analysis. Power calculations for sample comparisons were determined based on the comparisons of means/proportions using PASS statistical software. Analysis was performed using GraphPad PRISM (version 7.01; CA, USA), Minitab Statistical (version 17) and/or R Softwares. $P$ values for pairwise tests, or multiplicity-adjusted post-tests of selected pairs, are reported in the Figure Legends. p<0.05 was considered significant.

## Acknowledgements

We would like to thank the clinical coordinators at Emory University School of Medicine, especially Rincy Varughese, Cameron England, Rachel Safeek, Ramona Rai, and Clayton Carruth. The SCOPE cohort was supported the UCSF/Gladstone Institute of Virology and Immunology CFAR (P30 AI027763) and CFAR Network of Integrated Systems (R24 AI067039). Additional support was provided by the Delaney AIDS Research Enterprise (DARE; AI096109, A127966) and amfAR Institute for HIV Cure Research (amfAR 109301). This work was supported in part by the Bill and Melinda Gates Foundation and the Collaboration for AIDS Vaccine Discovery (BDW) and the Harvard University Center for AIDS Research grant P30 AI060354 (BDW), supported by the following NIH co-funding and participating Institutes and Centers: NIAID, NCI, NICHD, NHLBI, NIDA, NIMH, NIA, FIC, and OAR. This work was also supported by the following NIH grants: P30AI050409, U01 AA020790, U10 AA013566, and DP1DA036463. We thank Dr. Ned Landau of NYU Medical Center for kind gift plasmids. RES is a NIDA Avant Garde awardee.

## Additional information

### Funding

| Funder | Grant reference number | Author |
|---|---|---|
| National Institutes of Health | P30AI050409 | Vincent C Marconi |
| National Institutes of Health | U01 AA020790 | Vincent C Marconi |
| Bill and Melinda Gates Foundation | | Bruce D Walker |
| Harvard University Center for AIDS Research | P30 AI060354 | Bruce D Walker |
| The Collaboration for AIDS Vaccine Discovery (CAVD) | | Bruce D Walker |
| UCSF/Gladstone Institute of Virology and Immunology | P30 AI027763 | Steven Deeks |
| CFAR Network of Integrated Systems | R24 AI067039 | Steven Deeks |
| Delaney AIDS Research Enterprise | DARE; AI096109,A127966 | Steven Deeks |
| The amfAR Institute for HIV cure research | amfAR 109301 | Steven Deeks |

| National Institute on Drug Abuse | DP1DA036463 | Richard E Sutton |

The funders had no role in study design, data collection and interpretation, or the decision to submit the work for publication.

## Author contributions

Elena Gonzalo-Gil, Conceptualization, Data curation, Formal analysis, Investigation, Methodology, Writing—original draft, Writing—review and editing; Patrick B Rapuano, Data curation, Formal analysis, Methodology; Uchenna Ikediobi, Resources, Data curation, Methodology; Rebecca Leibowitz, Ayse K Coskun, Data curation, Methodology; Sameet Mehta, Software, Formal analysis; J Zachary Porterfield, Teagan D Lampkin, Vincent C Marconi, David Rimland, Bruce D Walker, Steven Deeks, Resources, Final approval of the version to be published; Richard E Sutton, Conceptualization, Resources, Supervision, Funding acquisition, Writing—review and editing

## Author ORCIDs

Elena Gonzalo-Gil http://orcid.org/0000-0002-0409-3094
Steven Deeks http://orcid.org/0000-0001-6371-747X
Richard E Sutton http://orcid.org/0000-0001-7418-2378

## Ethics

Human subjects: The study was approved by both the Yale IRB (Yale New Haven Hospital and other Yale-affiliated HIV clinics in Connecticut; IRB protocol HIC#1305012068), and the local IRBs (the SCOPE cohort from UCSF, the Ragon Institute of MGH, MIT and Harvard, and from Veterans Medical Center HIV clinics from Atlanta and Dallas) and informed, written consent was obtained from all subjects. All ethical guidelines regarding human subjects investigation were adhered to. None of the investigators had or currently have a real or perceived conflict of interest with regard to this work.

## Decision letter and Author response

Decision letter https://doi.org/10.7554/eLife.44360.025
Author response https://doi.org/10.7554/eLife.44360.026

# Additional files

## Supplementary files

• Supplementary file 1. Clinical characteristics of EC/VC cohort. Legend: F: female; M: male; F*: transgender male to female; No: number; Yr: year; ABC: Abacavir; ATV: Atazanavir; AZT: Zidovudine; COB: Cobicistat; DRV: Darunavir; DTG: Dolutegravir; EFV: Efavirenz; EGV: Elvitegravir; FTC: Emtricitabine; IFNA: Interferon alpha; LPV/r: Lopinavir/Ritonevir; MVC: Maraviroc; NVP: Nevirapine; PRED: Prednisolone; RGV: Raltegravir; RTV: Ritonavir; STRIBLD (EGV, COB, FTC, TDF); TAF: Tenofovir alafenamide; TDF: Tenofovir; 3TC: Lamivudine; IVDU: intravenous drug use; Homo: homosexual contact; Hetero: heterosexual contact; HCV: hepatitis C virus
DOI: https://doi.org/10.7554/eLife.44360.015

• Supplementary file 2. Significant genes obtained by RNA-seq analysis in CD4 +T cells from ATL2 family members.
DOI: https://doi.org/10.7554/eLife.44360.016

• Transparent reporting form
DOI: https://doi.org/10.7554/eLife.44360.017

## Data availability

Sequencing data have been deposited in GEO under accession code GSE122323. This SuperSerie is composed of the following SubSeries: GSE122321 (RNAseq) and GSE122322 (ATAC-seq). All data generated or analysed during this study are included in the manuscript and supporting files.

The following datasets were generated:

| Author(s) | Year | Dataset title | Dataset URL | Database and Identifier |
|-----------|------|---------------|-------------|-------------------------|
| Elena Gonzalo-Gil | 2018 | Transcriptional Down-regulation of CCR5 in a Subset of HIV+ Controllers (RNA-Seq) | https://www.ncbi.nlm.nih.gov/geo/query/acc.cgi?acc=GSE122321 | NCBI, GSE122321 |
| Elena Gonzalo-Gil | 2018 | Transcriptional Down-regulation of CCR5 in a Subset of HIV+ Controllers (ATAC-Seq) | https://www.ncbi.nlm.nih.gov/geo/query/acc.cgi?acc=GSE122322 | NCBI, GSE122322 |
| Elena Gonzalo-Gil | 2018 | Transcriptional Down-regulation of CCR5 in a Subset of HIV+ Controllers | https://www.ncbi.nlm.nih.gov/geo/query/acc.cgi?acc=GSE122323 | NCBI, GSE122323 |

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
