## [Decision Letter]

Thank you for submitting your article "Transcriptional down-regulation of *ccr5*in a subset of HIV+ controllers and their family members" for consideration by *eLife*. Your article has been reviewed by three peer reviewers, including Frank Kirchhoff as the Reviewing Editor and Reviewer #1, and the evaluation has been overseen by Michel Nussenzweig as the Senior Editor. The following individuals involved in review of your submission have agreed to reveal their identity: Felipe Diaz-Griffero (Reviewer #2).

The reviewers have discussed the reviews with one another and the Reviewing Editor has drafted this decision to help you prepare a revised submission.

Summary:

Some HIV-1-infected individuals are capable of efficiently controlling viral replication. The reasons for this are incompletely understood and most likely complex. In the present study, the authors analyzed a large group of 131 HIV-1 controllers. Major finds are: (i) CD4 T cells from a subset of ECs/VCs are relatively resistant to R5-tropic HIV infection in vitro, compared to those from other ECs/VCs or uninfected controls; (ii) the resistance phenotype is associated with lower levels of CCR5, likely due to reduced active transcription of CCR5; (iii) CD4+ T cells from ECr/VCr have highly accessible chromatin over ~500Kb region around CCR2 and CCR5, which does not seem to fit with decreased transcription of these genes and (iv) evidence from one VCr that the resistance phenotype is heritable.

Genetic determinants of effective control of HIV-1 are of significant interest. In the present study a large number of HIV-1 controllers was examined technologies to study an important question about heterogeneous mechanisms of natural virologic control. Altogether, the data are well presented and for most part convincing. However, as appreciated by the authors the study has some limitations; e.g. that no viremic group of HIV-1-infected individuals was analyzed for comparison. The results are of significant interest but several issues should be addressed prior to publication.

Essential revisions:

1) One concern is the initial selection approach. Substantial variation is also observed for CXCR4- and VSV-G-mediated HIV-1 infection of cells from different donors (Figure 1B). If this phenotype is stable (as confirmed for CCR5), selection of the most X4 and/or VSV-G resistant subgroup of individuals would also result in significant differences to other groups in subsequent analyses. Thus, the results clearly support that CD4+ T cells of some EC/VCs express relatively low levels of CCR5. Whether this occurs more frequently than in viremic individuals and was causative for control is difficult to assess and should be critically discussed.

2) A potential reason for the ECr/VCr phenotype not explored in the manuscript is that circulating CD4 T cells from ECr/VCr might be low in effector memory (EM) cells. Among CD3+CD4+ cells, the EM population can be defined by a CD45RO+/CD27- phenotype, with the CD45RO+/CD27+ population defined as central memory (CM). In blood, most CCR5+ cells (as well as MIP-1a- and -1b-producing cells) show an EM phenotype. EM cells may be more easily infected by R5-tropic viruses than are CM cells, and may also be more likely to express the virus genes when infected. If the ECr/VCr phenotype correlated with low frequencies of EM cells among total CD4 T cells, this could mean that the phenotype is not due to atypical gene transcription on a region of 3p21 in all CD4 T cells, but instead to an atypical pattern of memory CD4 T cell differentiation in which the most susceptible target cells arise less frequently. If samples are available, additional FACS analyses should be performed, incorporating the CD27 marker in their flow panel to distinguish effector memory and central memory T cells. This will also help to better assess whether T cell differentiation rather than transcription might be different. In either case, it seems that only CD45RO/RA was determined to define "effector memory" cells in the Results section; the term "memory" seems more appropriate here since effector memory represent the CD27- subset within the RA-RO+ population.

3) Susceptibility to HIV-1 was only tested and confirmed using single-round infection assays and Envs of two macrophage-tropic HIV-1 strains. The resistance of CD4+ T cells (and PBMCs) from some representative individuals should be confirmed using primary CCR5-tropic HIV-1 strains (and X4 controls) in a spreading infection.

4) As ATAC-seq and Rpb1-ChIP results suggest more accessible chromatin structure and more RNA polymerase occupancy around CCR2 and CCR5 in ECr/VCr donors, the down-regulation of CCR2/CCR5 mRNA is unexpected. Can the authors provide additional information or speculation to help the reader understand these seemingly conflicting findings? Would it be useful to check repressive chromatin markers like H3K27me3 or H3K9me3 to confirm that these levels decrease the ~500 Kb region around CCR2 and CCR5? Either additional experiments or simply additional explanation may be sufficient to address this issue before publication.

5) Figure 1. The authors should plot the infectivity for each of the 21 samples showing infection with VSV-G, R5 and X4 in the same plot (X4, R5, and VSV-G normalized viruses for p24). The group data looks fine, but the reader cannot appreciate the differences of infectivity when comparing VSV-G (that enters T cells) and R5 tropic viruses. This will also allow the investigator to detect the more extreme cases where the difference between VSV-G and R5 infection are the greatest and illustrate this difference in the wild type samples. The reader would like to know the magnitude of resistance for each of the 21 samples.

6) One question is whether HIV-1-YU2, HIV-ADA, and HIV-1-VSV-G were normalized for p24 before infection. This is important to understand the magnitude of the differences among samples.

7) Figure 3. This figure showed one of the most important findings of the paper and should be plotted per sample showing the variability in control cells (no EC/VC). CCR2 and CCR5 for each sample like in Figure 3C should be shown. Again, this could help to identify the strongest phenotypes.

---

## [Author Response]

Essential revisions:1) One concern is the initial selection approach. Substantial variation is also observed for CXCR4- and VSV-G-mediated HIV-1 infection of cells from different donors (Figure 1B). If this phenotype is stable (as confirmed for CCR5), selection of the most X4 and/or VSV-G resistant subgroup of individuals would also result in significant differences to other groups in subsequent analyses. Thus, the results clearly support that CD4+ T cells of some EC/VCs express relatively low levels of CCR5. Whether this occurs more frequently than in viremic individuals and was causative for control is difficult to assess and should be critically discussed.

We agree with the reviewer that there is variability in percentage of infected cells using X4-tropic virus and/or VSVG; this is unfortunately unavoidable when using primary T cells. However, this variability is observed in all groups (Figure 1B), including Controls (Ctrl), EC/VC, and ECr/VCr, those with the resistant phenotype to R5-tropic viruses. Based upon the results observed in the first experiment (Figure 1—figure supplement 1A and 1B), where we initially selected R5-tropic resistant samples, we did not observe samples with both R5-tropic and X4-tropic resistance (none of the% infected cells in EC/VC group were lower than the% infected cells in Ctrl for the X4-tropic virus used). Only UCSF-56 (see Figure 1—figure supplement 2A) showed resistance to R5-tropic with relative resistance to X4-tropic virus. To investigate whether individuals with R5-tropic resistance also had lower susceptibility to either X4-tropic or VSV G-pseudotyped virus, we correlated T cell susceptibility to R5-virus to that of X4 and also R5-virus to that of VSVG (see Figure 1—figure supplement 2B). The absence of a statistically significant correlation between R5 and X4 or R5 and VSVG infectivity suggest that activated T cells of these ECr/VCr are specifically resistant to R5-tropic virus, and susceptible to X4-tropic virus. Most importantly, these ECr/VCr T cells that are resistant to R5-tropic virus are susceptible to VSVG, used as a pan-tropic control, indicating that the R5-tropic resistance is not due to a lower ability of those T cells to be infected, rather that resistance is specific to R5-tropic virus. This information is now included in Results section and in a figure legend (Figure 1—figure supplement 2).

Additional evidence that these 21 T cell samples (ECr/VCr) are only resistant to R5-tropic, but susceptible to X4-tropic, virus is the fact that ccr2 and ccr5 mRNA levels were down-regulated in this group compared to Ctrl and remaining EC/VCs, with cxcr4 mRNA levels comparable between groups (Figure 2A). The fact that ccr5 mRNA levels positively correlated with the percentage of infected cells using R5-tropic virus (Figure 2B), coupled with the absence of correlation between ccr5 and cxcr4 mRNA levels, clearly support the idea that these ECr/VCr T cells were resistant only to R5-tropic and susceptible to X4-tropic and pan-tropic virus.

It is true that we cannot prove definitively that this R5-resistance phenotype is more prevalent in EC/VC compared to viremic individuals. There are three major reasons why we are unable to study HIV+ individuals who are viremic: (i) they are now quite rare in the U.S. and very difficult to recruit, especially prior to initiating therapy, (ii) it would be unethical for us to withhold ART in order to study them longitudinally (and longitudinal study is critical here to demonstrate stability and reproducibility of the phenotype, and (iii) it is very difficult to recover sufficient quantities of T cells from viremic individuals for these sorts of studies. We were able, however, to recruit 10 HIV+ progressors (Prog) on anti-retroviral therapy. Our results show that R5-resistance was only observed in a subset of EC/VC CD4+ T cells and not in those of Prog or Ctrl. We observed relative resistance to R5-tropic HIV in CD4+ T cells from ECr/VCr compared to remaining EC/VC, Ctrl, and Prog (Ctrl 1.05 ± 0.81%; Prog 0.87 ± 0.36%; EC/VC 1.09 ± 0.75%; ECr/VCr 0.20 ± 0.16%; P<0.0001). However, our results showed equal susceptibility to X4-tropic HIV (Ctrl 3.07 ± 1.32%; Prog 2.79 ± 1.62%; EC/VCs 3.64 ± 1.78%; ECr/VCr 2.96 ± 2.01%) and VSV G pseudoviral particles among the groups (Ctrl 34.8 ± 9.35%; Prog 37.7 ± 6.4%; EC/VCs 29.32 ± 11.71%; ECr/VCr 32.87 ± 10.08%; data not shown). We readily admit, however, that these experiments were performed in a small scale with only 10 Prog, and additional experiments should be performed to confirm these results.

2) A potential reason for the ECr/VCr phenotype not explored in the manuscript is that circulating CD4 T cells from ECr/VCr might be low in effector memory (EM) cells. Among CD3+CD4+ cells, the EM population can be defined by a CD45RO+/CD27- phenotype, with the CD45RO+/CD27+ population defined as central memory (CM). In blood, most CCR5+ cells (as well as MIP-1a- and -1b-producing cells) show an EM phenotype. EM cells may be more easily infected by R5-tropic viruses than are CM cells, and may also be more likely to express the virus genes when infected. If the ECr/VCr phenotype correlated with low frequencies of EM cells among total CD4 T cells, this could mean that the phenotype is not due to atypical gene transcription on a region of 3p21 in all CD4 T cells, but instead to an atypical pattern of memory CD4 T cell differentiation in which the most susceptible target cells arise less frequently. If samples are available, additional FACS analyses should be performed, incorporating the CD27 marker in their flow panel to distinguish effector memory and central memory T cells. This will also help to better assess whether T cell differentiation rather than transcription might be different. In either case, it seems that only CD45RO/RA was determined to define "effector memory" cells in the Results section; the term "memory" seems more appropriate here since effector memory represent the CD27- subset within the RA-RO+ population.

As suggested, we analyzed additional T cell samples from a subset of Ctrl, EC/VC, and ECr/VCr to quantify the percentage of circulating Effector Memory (EM) vs Central Memory (CM) T cells and corresponding levels of CCR5 in both populations. These results are now included in Figure 3D (see also Figure 3 legend), in the Results section, and in the Materials and methods section. Overall, the percentage of EM (CD45RO+CD27-) cells in ECr/VCr was higher than in Ctrls and EC/VCs, and CM (CD45RO+CD27+) cells being relatively lower in ECr/VCr than the other two groups (but these differences were not significant). In general, and as we expected, the percentage of EM cells and CCR5+ cells with EM phenotype were higher than that of CM cells. More interestingly, the percentages of CCR5+ cells were reduced in both EM and CM cells in ECr/VCr compared to the other two groups (Ctrl and remaining EC/VC), indicating that CCR5 levels were lower in ECr/VCr not only in EM cells, the cells that are more susceptible in general to R5-tropic virus, but also in CM cells, T cells that are typically less susceptible to R5-tropic infection. Taken together, these data suggest a transcriptional mechanism of ccr5 down-regulation associated with R5-tropic resistance in a subset of EC/VC, not a difference in EM and CM T cell populations in ECr/VCr.

In addition, the term “effector memory” has been changed to “memory” accordingly (Results section and Figure 3 legend).

3) Susceptibility to HIV-1 was only tested and confirmed using single-round infection assays and Envs of two macrophage-tropic HIV-1 strains. The resistance of CD4+ T cells (and PBMCs) from some representative individuals should be confirmed using primary CCR5-tropic HIV-1 strains (and X4 controls) in a spreading infection.

We agree that it is of interest to quantify viral kinetics using replication-competent virus. We have now included experiments with replication-competent virus using R5-tropic pNL-BaL and X4-tropic HIV-NL4-3ΔR1, showing replication curves in CD4+ T cells from representative Ctrl, EC/VC, and ECr/VCr (n=2 per group tested in triplicate, see Figure 1—figure supplement 2C). This figure demonstrates relative resistance to and reduced replication of X4- and R5-tropic virus in CD4+ T cells of ECr/VCr, compared to those of EC/VC and Ctrl, over the 21 days analyzed (see quantitation of AUC, or area under the curve). Most interestingly, infection using X4-tropic viruses demonstrated reduced infection in all EC/VC (with or without the R5-resistance phenotype by single-cycle assay), with virtual absence of replication in ECr/VCr. Importantly, this lack of replication was not associated with any cytotoxicity, as all CD4+ T cells at the end of the 3-week period were very healthy and viable, with no cell death evident. We wish to point out that the absence of differences in replication at day 3 post-infection can be explained by the fact that in the initial transfection of 293T cells to produce virus we included VSV G expression plasmid together with plasmids encoding full-length X4- or R5-tropic virus. Inclusion of the VSV G plasmid was necessary to initiate infection of the CD4+ T cells with multicycle virus (otherwise, the T cells were extremely poorly infected, at least in our hands). These data suggest relative resistance to X4-tropic virus in activated CD4+ T cells of EC/VC and suggest a more complex mechanism for ECr/VCr, other than simply down-regulation of CCR5. That single cycle X4 infectivity is normal in ECr/VCr T cells suggests an additional late block to viral replication, which is the subject of ongoing investigation. A paragraph has been included in the Results section and in the Materials and methods section.

Primary viral isolates (e.g., transmitted/founder strains) were not tested here. There is no evidence in the literature that these viruses use a co-receptor other than ccr5, and the more important conclusion of our paper is that the reason that ECr/VCr T cells are relatively resistant to R5 virus is due to RNA down-regulation of ccr5 (and not CD4), which appears to be genetic in nature. Testing PBMCs would only muddy the waters since it is a mixed cell population (especially since inhibitory CD8+ T cells will be present); we performed all single cycle experiments using highly purified CD4+ T cells. Furthermore, for most of these subjects CD4+ T cells and not PBMCs were available; thus, we performed the experiments with replication-competent virus using activated CD4+ T cells instead of PBMCs. Parenthetically, if one treats PBMCs with anti-CD3/CD28 in the presence of IL2 after a few days the cell population typically ends up being mostly CD4+/CD8+ T cells.

4) As ATAC-seq and Rpb1-ChIP results suggest more accessible chromatin structure and more RNA polymerase occupancy around CCR2 and CCR5 in ECr/VCr donors, the down-regulation of CCR2/CCR5 mRNA is unexpected. Can the authors provide additional information or speculation to help the reader understand these seemingly conflicting findings? Would it be useful to check repressive chromatin markers like H3K27me3 or H3K9me3 to confirm that these levels decrease the ~500 Kb region around CCR2 and CCR5? Either additional experiments or simply additional explanation may be sufficient to address this issue before publication.

As mentioned, it is very intriguing that the observed down-regulation of ccr2 and ccr5 and lower active transcription as measured by ccr5 and ccr2 levels after immunoprecipitation using anti-Rpb1 is associated with more accessible chromatin in ECr/VCr T cells compared to that of Ctrl and remaining EC/VC. Since our results performed using anti-H3K4me3 suggest more activated chromatin markers in ECr/VCr compared to other two groups, it would be interesting to analyze repressive chromatin markers such as H3K9me3 or H3K27me3. Unfortunately, the samples were very limiting, and this could not be performed. ChIP qPCR experiments require at least 2 million CD4+ T cells and it would be necessary to perform the experiment multiple times with enough numbers of subjects per group to be confident in the results. As mentioned in the Discussion section, H3K4Me3 has been previously associated with open chromatin, correlating with higher levels of transcripts (Heintzman et al., 2007, Bernstein et el., 2005). Since our results suggest that more open chromatin over ~500Kb region on chr3, including surrounding ccr2 and ccr5, is associated with lower transcription of both genes in ECr/VCr, it is expected that repressive chromatin markers would also be decreased, despite the down-regulation of multiple genes on 3p21. We can only speculate why this is so, but one possibility is that ECr/VCr CD4+ T cells express a dominant negative transcription factor (i.e., a DN STAT, as intimated in the Discussion section), such that the cell overcompensates for reduced transcription in this region by somehow attempting to keep the chromatin even more open. This would also explain the 5-10 fold RNA down-regulation (presumably affecting alleles or genes on both chromosomes) and the fact that the R5-resistance phenotype appears to be autosomal dominant in inheritance pattern. Despite some investigation, we do not have any direct evidence for a DN transcription factor—this is also the subject of future work—and thus are loathe to include this degree of speculation in the Discussion section.

5) Figure 1. The authors should plot the infectivity for each of the 21 samples showing infection with VSV-G, R5 and X4 in the same plot (X4, R5, and VSV-G normalized viruses for p24). The group data looks fine, but the reader cannot appreciate the differences of infectivity when comparing VSV-G (that enters T cells) and R5 tropic viruses. This will also allow the investigator to detect the more extreme cases where the difference between VSV-G and R5 infection are the greatest and illustrate this difference in the wild type samples. The reader would like to know the magnitude of resistance for each of the 21 samples.

As requested, we have now included plots showing single-cycle infectivity for all 21 ECr/VCr against X4, R5, and VSVG pseudotyped viruses in the same plot (Figure 1—figure supplement 2A). Also, a figure showing the correlation between R5 and X4 and R5 and VSVG for all 21 ECr/VCr is also now included (see Figure 1—figure supplement 2B). A sentence has been added in the Results section. The absence of a statistically significant correlation between R5 and X4 and R5 and VSVG susceptibility suggest that these ECr/VCr are specifically resistant to R5-tropic virus, being susceptible to X4-tropic virus (note that only UCSF56 showed relative lower percentages of infectivity against R5 and X4-tropic viruses—we would like to study that subject further but have been unable to obtain more PBMCs). And, as stated above, these ECr/VCr T cells are resistant to R5-tropic viruses independent of VSVG susceptibility, indicating that the R5-tropic resistance is not due to a lower ability to be infected, due to poor cell viability or activation status, for example.

For normalization of the amount of virus used, we used infectivity on GHOST cell derivatives by performing a titration of each virus using GHOST.HI5 (for R5-tropic virus) or GHOST.X4 (for X4-tropic and VSVG pan-tropic virus) cells. Based upon the end-point titration by flow cytometry, we determined the Infectious Units per ml (IU/ml) for each lot of virus and the same quantity of IU was used to infect CD4+ T cells of the subjects (Ctrl, EC/VC, and ECr/VCr). Critically, the same volumes and tissue culture plate format were also used for the infectivity assays. With regards to normalization of the multicycle virus, this was performed on TZMbls, and the same quantity of virus was used for each of the cell samples. A paragraph has been included in the Materials and methods section to clarify how the normalization was performed.

6) One question is whether HIV-1-YU2, HIV-ADA, and HIV-1-VSV-G were normalized for p24 before infection. This is important to understand the magnitude of the differences among samples.

Please refer to our response above regarding normalization using IU based upon titering in GHOST reporter cell lines, not by p24 CA.

7) Figure 3. This figure showed one of the most important findings of the paper and should be plotted per sample showing the variability in control cells (no EC/VC). CCR2 and CCR5 for each sample like in Figure 3C should be shown. Again, this could help to identify the strongest phenotypes.

As requested, we have included individual plots for all 21 EC/VC with the resistance phenotype for CCR2 (Figure 3—figure supplement 1) and CCR5 expression (Figure 3—figure supplement 2), as well as individual plots for Ctrls (normal healthy donors) for comparison. This information has been included in the Results section. Also, a graph showing the positive correlation between% ccr2 and% ccr5 has been also included (see Figure 3F) and the corresponding information has been included in the Results section and in Figure 3 legend.